JCB Journal of Cell Biology

# Phosphatidic acid drives spatiotemporal distribution of Pex30 at ER-LD contact sites

Morgan House[1], Karan Khadayat[1], Thomas N. Trybala[2], Nikhil Nambiar[3], Elizabeth Jones[1], Steven M. Abel[3], Joshua Baccile[2], and Amit S. Joshi[1]

Lipid droplets (LDs) are ubiquitous neutral lipid storage organelles that form at discrete subdomains in the ER bilayer. The assembly of these ER subdomains and the mechanism by which proteins are recruited to them is poorly understood. Here, we investigate the spatiotemporal distribution of Pex30 at the ER-LD membrane contact sites (MCSs). Pex30, an ER membrane–shaping protein, has a reticulon homology domain, a dysferlin (DysF) domain, and a Duf4196 domain. Deletion of *SEI1*, which codes for seipin, a highly conserved protein required for LD biogenesis, results in accumulation of Pex30 and phosphatidic acid (PA) at ER-LD contact sites. We show that PA recruits Pex30 at ER subdomains by binding to the DysF domain. The distribution of Pex30 as well as PA is also affected by phosphatidylcholine (PC) levels. We propose that PA regulates the spatiotemporal distribution of Pex30 at ER subdomains that plays a critical role in driving the formation of LDs in the ER membrane.

## Introduction

Lipid droplets (LDs) are neutral lipid storage organelles that form in the ER bilayer. LDs have a unique architecture as the core of a LD contains neutral lipids, such as triacylglycerol (TAG) and sterol esters, that are surrounded by an amphipathic phospholipid monolayer. LDs are highly dynamic organelles that form, grow, and shrink in size and number in response to cellular cues (Olzmann and Carvalho, 2019; Kumari et al., 2023; Mathiowetz and Olzmann, 2024). LD biogenesis begins with accumulation of neutral lipids in the ER bilayer to form a lens-like structure (Thiam and Ikonen, 2021; Choudhary et al., 2015). The lens-like structure grows to bud toward the cytosolic face of the ER membrane (Deslandes et al., 2017). In *Saccharomyces cerevisiae*, the LDs stay connected to the ER membrane, whereas in mammalian cells they might detach from the ER membrane (Jacquier et al., 2011). Formation of LDs can be influenced by several factors, such as ER membrane lipid composition, membrane curvature, surface tension, and proteins localized to these ER subdomains (Choudhary et al., 2018; Thiam and Forêt, 2016; Santinho et al., 2020; Ben M'barek et al., 2017). In *S. cerevisiae*, several proteins have been demonstrated to localize at sites of nascent LD formation. These include proteins such as Nem1, Pah1, and Spo7, required for DAG formation; seipin (Sei1), localized at ER-LD contact sites and traps TAG; Pex30, an ER membrane–tubulating protein; and additional biogenesis proteins such as Yft2, Pet10, and Erg6. However, assembly of LD biogenesis sites is poorly understood (Choudhary et al., 2015;

Choudhary et al., 2020; Joshi et al., 2018; Karanasios et al., 2013; Wolinski et al., 2015; Wang et al., 2014; Gao et al., 2017; Arlt et al., 2022; Zoni et al., 2021).

Seipin is an ER membrane protein that plays a crucial role in LD biogenesis. This highly conserved protein is encoded by *BSCL2* in humans, and mutations in this gene have been shown to cause severe lipodystrophy (Magré et al., 2001; Li et al., 2022). Loss of seipin leads to fewer and larger LDs (Fei et al., 2008). The structure of seipin revealed that the core elements of the protein in fly, yeast, and humans form large oligomeric complexes comprising of 12, 10, and 11 monomeric subunits, respectively. The hydrophobic helices induce TAG nucleation, whereas transmembrane domains affect LD maturation (Arlt et al., 2022; Kim et al., 2022; Klug et al., 2021). Other proteins, such as LDAF1 (related to yeast Ldo16 and Ldo45) in mammals and Ldb16 in yeast, play a crucial role along with seipin in LD nucleation, as absence of either of these proteins leads to defects in TAG nucleation and LD morphology (Chung et al., 2019; Teixeira et al., 2018; Eisenberg-Bord et al., 2018; Prasanna et al., 2021). Seipin also consists of two β-sheets, each containing four antiparallel β-strands, which are similar to lipid-binding C2 domains. In vitro studies show that the full-length as well as the luminal domain of seipin binds to the anionic phospholipid phosphatidic acid (PA). In addition to this, the full-length seipin can also bind to phosphoinositol-3-phosphate (PI3P) (Yan et al., 2018). PI3P was recently shown to localize at LD biogenesis sites in the ER

[1]Department of Biochemistry & Cellular and Molecular Biology, University of Tennessee, Knoxville, TN, USA; [2]Department of Chemistry, University of Tennessee, Knoxville, TN, USA; [3]Department of Chemical and Biomolecular Engineering, University of Tennessee, Knoxville, TN, USA.

Correspondence to Amit S. Joshi: ajoshi18@utk.edu.

membrane. Additionally, decreasing PI3P levels can rescue the LD phenotype of seipin mutant cells (Lukmantara et al., 2022). Furthermore, loss of seipin leads to accumulation of PA at ER-LD contact sites, suggesting it regulates ectopic accumulation of PA in the ER membrane (Han et al., 2015; Wolinski et al., 2015).

Previously, we showed that Pex30 is a reticulon-like ER membrane–shaping protein (Joshi et al., 2016). Pex30 has three domains, a reticulon homology domain (RHD), a dysferlin (DysF) domain, and a Duf4196 domain. Endogenous Pex30 localizes to multiple ER subdomains, forming ~20 puncta per cell (Joshi et al., 2016; Ferreira and Carvalho, 2021). Some of these subdomains are the sites of peroxisome and LD biogenesis (Joshi et al., 2016; Joshi et al., 2018; Wang et al., 2018). Pex30 colocalizes with Nem1, seipin, and an ER-DAG sensor at LD biogenesis sites (Joshi et al., 2018). While Pex30 is mainly associated with cytoplasmic LDs, it is not enriched with nuclear LDs (Romanauska et al., 2024; Joshi et al., 2018). Pex30 and the Pex30-like proteins Pex28, Pex29, Pex31, and Pex32 localize to multiple membrane contact sites (MCSs). Different Pex30 complexes function at distinct MCSs (Ferreira and Carvalho, 2021). In addition to the ER-peroxisome contact site where Pex30 is bound to Pex28 and Pex32, Pex30 also complexes with Pex29 at the nuclear-vacuolar junction. The Pex30 RHD is essential for ER membrane tubulation and also interacts with Pex30-like proteins, whereas the DysF domain is essential to regulate peroxisome number and localization of Pex30 at the nuclear-vacuolar junction (Deori et al., 2023; Ferreira and Carvalho, 2021). However, the specific function of the DysF domain is not known. Pex30 phosphorylation also regulates peroxisome abundance (Deori et al., 2022). In addition to peroxisomes, loss of Pex30 leads to delayed formation of new LDs, possibly due to delayed recruitment of the Lro1 enzyme required for TAG synthesis (Choudhary et al., 2020; Joshi et al., 2018). Cells devoid of seipin and Pex30 exhibit severe growth defects, suggesting that the function of Pex30 is vital in the sei1Δ mutant (Joshi et al., 2018; Wang et al., 2018). The sei1pex30Δ mutant also exhibits an increase in total cellular levels of phosphatidylcholine (PC), phosphatidylinositol (PI), DAG, and TAG, as well as increase in ER membrane proliferation and a severe defect in LD morphology; LDs in the sei1pex30Δ mutant are highly clustered, big as well as small, and are entangled in the ER membrane (Wang et al., 2018; Joshi et al., 2018). These results indicate that both Pex30 and seipin are required for assembly of ER subdomains associated with LD biogenesis. Here, we investigate how the spatiotemporal distribution of Pex30 is regulated at ER subdomains associated with LD biogenesis.

Previous studies have shown that loss of seipin in S. cerevisiae alters the distribution of Pex30 in the ER membrane. In WT cells, Pex30 is distributed in multiple puncta in the ER membrane; however, in the sei1Δ mutant, Pex30 shows fewer puncta as it accumulates into a large punctum associated with LD biogenesis sites (Joshi et al., 2018). Here, we utilize this observation to investigate the factors that regulate the distribution of Pex30 at ER subdomains. We show that Pex30 accumulates at ER-LD contact sites which are also enriched with PA. Also, we find that presence of LDs and DysF domain is essential for Pex30 accumulation at the ER-LD contact sites. In vitro as well as in silico studies reveal that the DysF domain binds PA. Thus, we propose that PA binds to the DysF domain to recruit Pex30 at the ER subdomains that drives LD biogenesis.

## Results

### Pex30 and PA accumulate at the ER-LD contact sites in sei1Δ

In WT cells, Pex30 punctae are localized at ER subdomains distributed throughout the ER membrane. Some of these punctae are associated with the LDs (Joshi et al., 2018). However, in the sei1Δ mutant, Pex30 accumulates at fewer ER subdomains than in WT cells to form supersized punctae (Fig. 1 A). Consistent with previous findings, we do not find significant increase in Pex30-GFP expression in the sei1Δ mutant (Fig. S1, A and B) (Wang et al., 2018). These Pex30 puncta colocalize with Nem1 and are associated with LDs (Joshi et al., 2018). To confirm that Pex30 is in the ER membrane and not on the LD surface, we used high-resolution Airyscan microscopy to visualize Pex30 localization in the sei1Δ mutant cells. In WT cells, Pex30-2xmCherry colocalizes with the ER membrane protein Sec63-GFP and is associated with LDs stained with monodansylpentane (MDH) dye. However, in sei1Δ cells, Pex30-2xmCherry accumulates to form much larger punctae that colocalize with Sec63-GFP and LDs, suggesting Pex30-2xmCherry is predominantly located at the ER-LD contact sites (Fig. 1, A and B). In the sei1Δ mutant, PA accumulates ectopically at the ER-LD contact sites (Wolinski et al., 2015; Han et al., 2015). To determine whether Pex30 and PA accumulate at the same ER-LD contact sites, we checked the localization of endogenously expressed Opi1-GFP, which binds PA (Loewen et al., 2004; Hofbauer et al., 2018) and Pex30-2xmCherry in the sei1Δ mutant. We found that Opi1-GFP punctae colocalized with Pex30-2xmCherry, suggesting Pex30 subdomains are enriched with PA (Fig. 1 C). Pex30-2xmCherry accumulation is not Opi1 dependent, as Pex30-2xmCherry also accumulates in sei1opi1Δ (Fig. S1 C). Moreover, we found that the ER tubule-forming reticulon protein Rtn1-GFP does not accumulate at ER-LD contact sites, indicating that the localization of Pex30-2xmCherry is specific to reticulon-like Pex30 (Fig. S1 D).

To determine whether Pex30 is required for ectopic accumulation of PA at ER-LD contact sites in the sei1Δ mutant, we measured Opi1-GFP distribution in the sei1pex30Δ mutant. As previously reported, Opi1-GFP localized to the nuclear membrane in WT cells and accumulated as punctae in the sei1Δ mutant cells. To check if Opi1 puncta are nuclear, we co-expressed Sec63-GFP and Opi1-mCherry in the sei1Δ mutant. We found that majority of the puncta accumulate on the nuclear and ER membrane, with very few puncta inside the nucleus (Fig. S1, E and F). While Opi1-GFP localization is WT-like in the pex30Δ mutant, it forms punctae in the sei1pex30Δ as in sei1Δ (Fig. 1, D and E). Interestingly, in sei1pex30Δ, the number of Opi1-GFP puncta per cell is significantly higher than in sei1Δ cells (Fig. 1F). Also, Opi1-GFP puncta in sei1pex30Δ cells were smaller in size than in sei1Δ (Fig. 1 D). It is possible that Pex30 generates local membrane curvature that sequesters PA at the ER-LD contact sites. It could also affect cellular PA levels. Indeed, we found a significant increase in cellular PA levels in sei1pex30Δ cells as compared with WT (Fig. 1 G). Furthermore, we found a

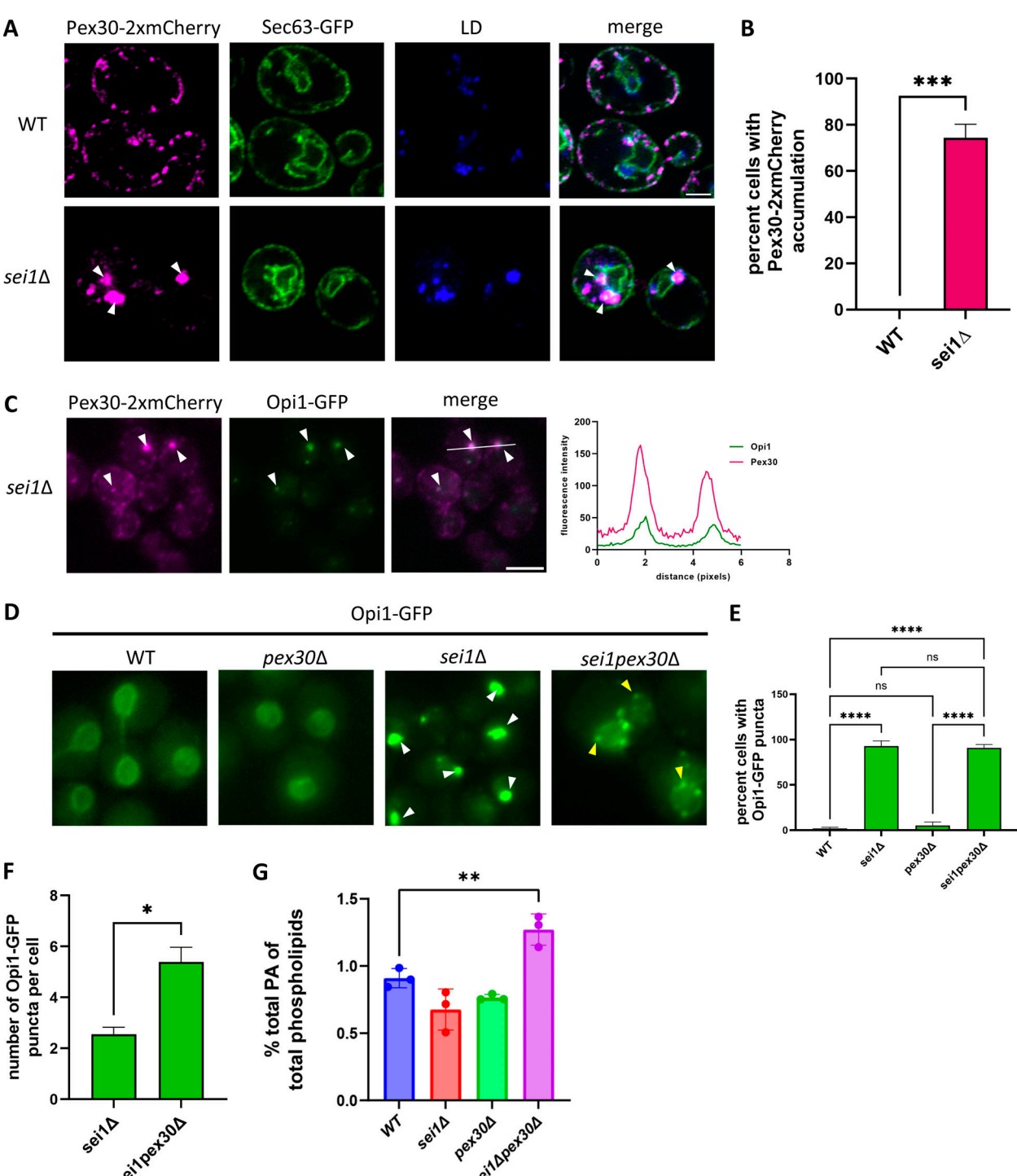

Figure 1. **Pex30 and PA accumulate at ER-LD contact sites in *sei1Δ*. (A)** Airyscan images (AS) of WT and *sei1Δ* cells endogenously expressing Pex30-2xmCherry and Sec63-GFP, an ER marker, on a plasmid. Cells were stained for LDs with MDH and imaged in stationary phase. White arrowheads denote Pex30-2xmCherry and LD puncta that colocalize in *sei1Δ*. Bar = 2 µm. **(B)** Quantification of Pex30-2xmCherry accumulation in WT versus *sei1Δ*. Bars show mean from three independent experiments and standard SEM. 100 cells per genotype from each replicate were analyzed and compared using an unpaired *t* test (***P < 0.0001). **(C)** Widefield images (WF) of *sei1Δ* cells endogenously expressing Pex30-2xmCherry and Opi1-GFP, a sensor for PA, in logarithmic phase. White arrowheads show Pex30-2xmCherry and Opi1-GFP puncta that colocalize; graph to the right of C shows signal intensity on the white line. Bar = 4 µm. **(D)** WF images of the indicated strains expressing Opi1-GFP on a plasmid in logarithmic phase. White arrowheads show large Opi1-GFP puncta in *sei1Δ*, and yellow arrowheads show small Opi1-GFP puncta in *sei1pex30Δ*. Bar = 4 µm. **(E)** Quantification of the percentage of cells showing Opi1-GFP puncta shown in D. Bars show mean from three independent experiments and SEM. 100 cells per genotype from each replicate were analyzed and compared using one-way ANOVA and Tukey's multiple comparison test (****P < 0.0001). **(F)** Quantification of the number of Opi1-GFP puncta in *sei1Δ* and *sei1pex30Δ* from cells shown in D. Bars show mean from three independent experiments and SEM. 100 cells per genotype from each replicate were analyzed and compared using an unpaired *t* test (*P < 0.05). **(G)** Phospholipid measurements of indicated strains by liquid-chromatography high-resolution mass spectrometry (LC-HRMS) of cell pellets (*n* = 3). Amount of total quantitated PA relative to total quantitated phospholipids measured in indicated strains (**P < 0.01).

significant increase in PA (16:0, 18:1) which is one of the most abundant PA species (Fig. S1 G) (Ganesan et al., 2016). Thus, Pex30 affects total cellular PA levels as well as PA distribution in the ER membrane.

## Pex30 accumulates at the ER-LD contact site after new LD formation and PA enrichment

To determine if PA recruits Pex30 at the ER-LD contact sites in the *sei1Δ* mutant, we used cells that do not exhibit PA accumulation. As shown previously, the ectopic accumulation of PA in the ER membrane in *sei1Δ* can be decreased if the cells do not contain any LDs. It was reported that, in cells devoid of seipin, PA enriches at the ER-LD contact sites only after formation of new SE only or TAG only LDs (Han et al., 2015). In *S. cerevisiae*, four enzymes produce neutral lipids—Are1 and Are2 generate SE, and Lro1 and Dga1 synthesize TAG. Thus, cells that lack all four enzymes do not form LDs (Jacquier et al., 2011). We used a strain devoid of Are1, Are2, Dga1, as well as seipin proteins, and expression of *LRO1* was regulated under the *GAL1* promoter. The strain also expressed Pex30-2xmCherry and Opi1-GFP, whereas LDs were stained with MDH. Cells lack LDs when grown in medium containing raffinose but form new LDs when switched to medium containing galactose. Before addition of galactose, cells have no LDs, and Opi1-GFP was uniformly localized to the nuclear membrane indicating no PA accumulation (Fig. 2 A) (Wolinski et al., 2015). Interestingly, Pex30-2xmCherry distribution was WT-like in this strain and does not accumulate as in the *sei1Δ* mutant (Fig. 1 A). Thus, LD formation is essential for accumulation of PA and Pex30-2xmCherry in cells devoid of seipin. As previously shown, after addition of galactose, we found that new LDs form at Pex30-2xmCherry puncta in 1 h (Joshi et al., 2018) (Fig. 2, A–C). This was followed by Opi1-GFP punctae formation in 40% of cells within 2 h indicating ectopic PA accumulation (Fig. 2 B). The Pex30-2xmCherry puncta accumulate to form supersized punctae at the ER-LD contact sites after 75% of the cells exhibit LD and Opi1-GFP puncta formation, suggesting ectopic PA accumulation precedes Pex30-2xmCherry accumulation at the ER subdomains where LDs form (Fig. 2 A–C). LD formation is inefficient upon PA accumulation in the ER membrane (Ben M'barek et al., 2017). Pex30 recruitment after PA accumulation at ER-LD contact sites could possibly favor LD budding. Thus, our findings indicate that Pex30 accumulation at the ER-LD contact site is a consequence of PA accumulation, possibly to generate membrane curvature at ER subdomains and maintain surface tension.

## DysF domain is required for recruitment of Pex30 at ER-LD contact sites

To determine which domain of Pex30 is essential for targeting it to ER-LD contact sites that are enriched with PA in the *sei1Δ* mutant, we generated several truncations of Pex30 tagged with GFP (Fig. 3 A). Only full-length Pex30-GFP and Pex30 (DUFΔ)–GFP accumulated at the ER-LD contact sites with Opi1-mCherry (Fig. 3, B and C). As expected, the Pex30 (RHDΔ)–GFP did not localize to the ER membrane and was mostly cytosolic (Fig. 3 B). Additionally, Pex30 (RHDΔ)–GFP exhibited a significant decrease in Opi1-mCherry puncta (Fig. 3 D). This suggests that the

Pex30 RHD plays a role in sequestering PA at the ER-LD contact sites. In some cells, Pex30(RHDΔ)–GFP is targeted to membranes, suggesting its ability to bind lipids (data not shown). In contrast, Pex30(DysFΔ)–GFP localizes to the ER membrane but fails to accumulate at the ER-LD contact sites with Opi1-mCherry, indicating that DysF domain is essential for recruiting Pex30 to ER-LD contact sites (Fig. 3, B and C). Next, we checked which of these domains of Pex30 are functionally indispensable. Both the RHD and the DysF domains are essential for Pex30 function, as the Pex30(RHDΔ)–GFP and Pex30(DysFΔ)–GFP plasmids did not rescue the growth defect of the *sei1pex30Δ* mutant (Fig. 3 E). Loss of function of Pex30 (RHDΔ)–GFP and Pex30 (DysFΔ)–GFP was not due to decreased expression, as there was no significant change in expression compared with Pex30-GFP (Fig. 3, F and G). Thus, our findings suggest that Pex30 RHD might be required for enriching PA at the ER-LD contact sites by providing local membrane curvature, and the DysF domain possibly recruits Pex30 to ER subdomains by interacting with PA.

## DysF domain binds PA

In *S. cerevisiae*, Pex30, Pex30-like proteins, and Spo73 harbor DysF domains (Okumura et al., 2015). In humans, myoferlin, DysF, and tectonin beta-propeller repeat containing 1 proteins contain the DysF domains and are implicated in membrane remodeling (Sula et al., 2014). The pathogenic mutations in dysferlinopathies, an autosomal recessive late onset progressive muscular dystrophy, are associated mainly with mutation in the DysF domain. DysF, a member of the ferlin family of proteins, is a membrane-anchored protein with seven calcium-dependent phospholipid-binding C2 domains, three ferlin domains, and two DysF domains, one nested inside the other (Strehle, 2008). The inner DysF domain of human DysF protein has several amino acid residues conserved with yeast proteins containing the DysF domain. The crystal structure of human inner DysF domain indicates the presence of two long antiparallel β-strands (β1 and β6), one at each terminus. The structure reveals arginine/tryptophan stacking in this domain predicted to have interactions with other proteins (Sula et al., 2014). However, the function of DysF domain remains unknown. Considering we found interaction of the Pex30(RHDΔ)–GFP with membranes and the role of DysF in recruiting Pex30 to PA-enriched subdomains (Fig. 3 B), we investigated if the DysF domain interacts with membrane phospholipids, specifically PA. To test this, we first purified the soluble DysF domain containing a hexahistidine (6xHis) tag using Ni-NTA affinity chromatography, followed by size exclusion chromatography (Fig. S2, A–C). Next, we tested the binding of the DysF domain with multiple lipids using a protein-lipid overlay assay. We found the DysF domain weakly binds not only PA but also PI3P, PI4P, and PI4,5P2 (Fig. S2 D). To determine if PI3P is enriched at Pex30 subdomains, cells endogenously expressing Pex30-2xmCherry and FYVE-GFP that bind PI3P (Gaullier et al., 1998) on a plasmid were stained with MDH for LDs and imaged. Pex30-2xmCherry associated with FVYE-GFP punctae and LDs, suggesting some Pex30 ER subdomains are enriched with PI3P in WT and *sei1Δ* (Fig. S2 E). Our findings are consistent with previous

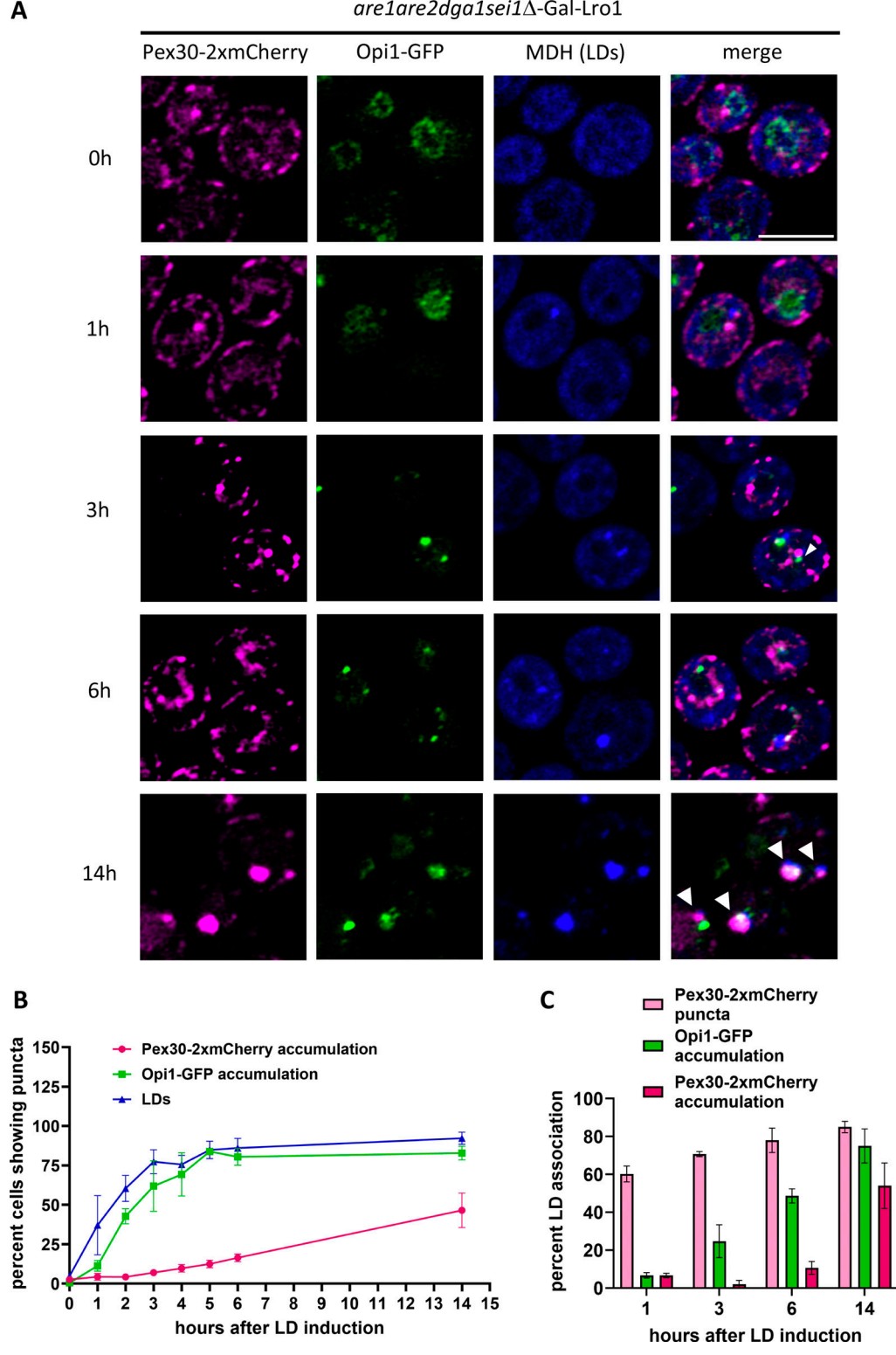

Figure 2. **PA accumulation at LD biogenesis sites precedes Pex30 accumulation. (A)** AS images of *are1are2dga1sei1Δ-GAL1-LRO1* cells endogenously expressing Pex30-2xmCherry and Opi1-GFP on a plasmid. Cells were stained for LDs with MDH and imaged at indicated time intervals after LD induction. Bar = 4 μm. **(B)** Quantification of experiment in A showing the percent of cells with Pex30-2xmCherry accumulation, Opi1-GFP accumulation, and LDs over time. Time points represent the mean of three independent experiments. 100 cells per time point from each replicate were analyzed. Error bars represent SEM. **(C)** Quantification of experiment in A showing LD association with Pex30-2xmCherry puncta, Opi1-GFP accumulation, and Pex30-2xmCherry accumulation over time from three independent experiments. 50 LDs per time point from each replicate were analyzed and bars show the mean and SEM. AS, Airyscan images.

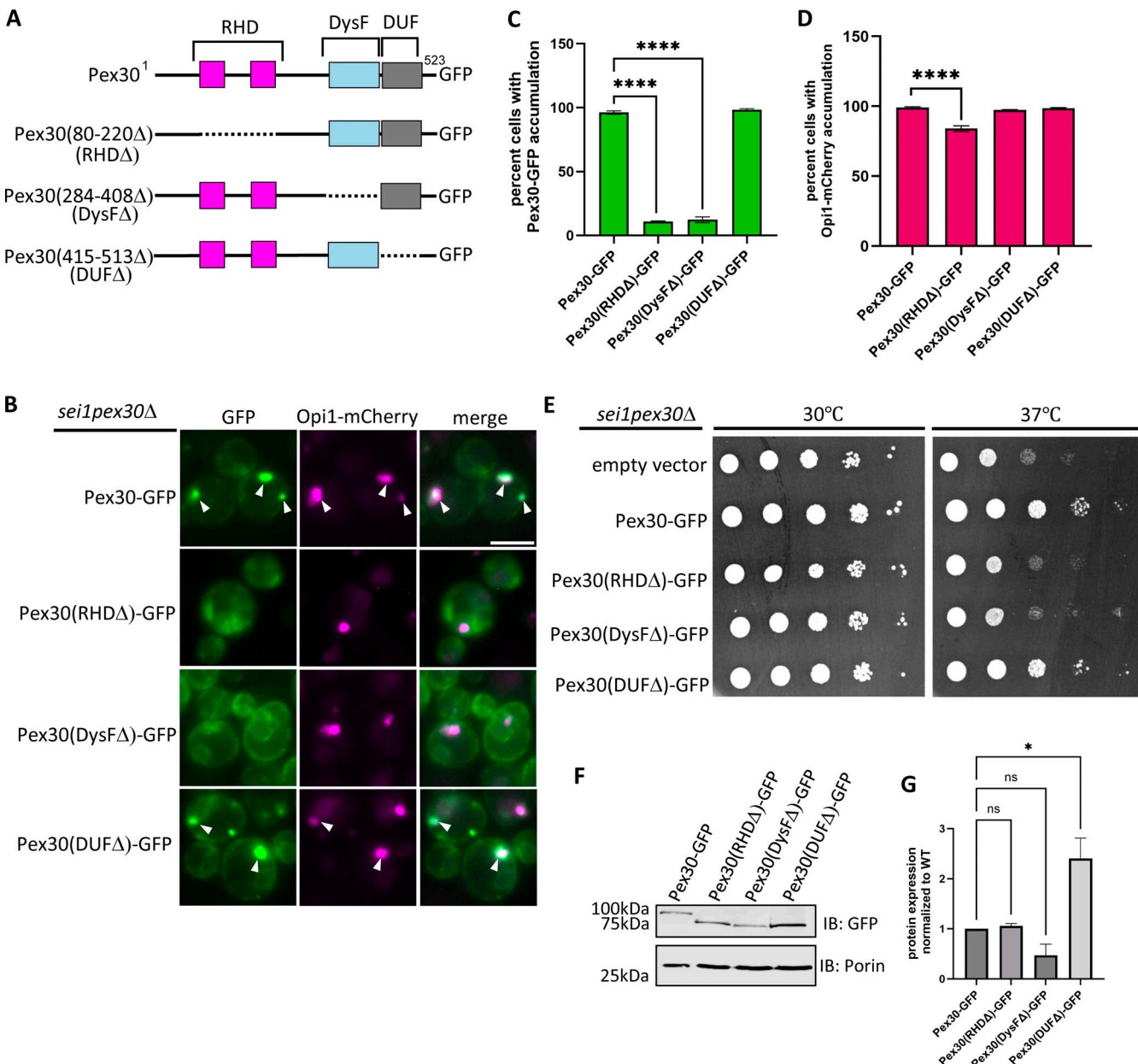

Figure 3. **DysF domain is required for recruitment of Pex30 at ER-LD contact sites. (A)** Schematic of Pex30 truncations tagged with GFP. **(B)** WF images of *sei1pex30Δ* cells endogenously expressing Opi1-mCherry and truncations of Pex30 tagged with GFP on a plasmid grown to logarithmic phase. White arrowheads denote Pex30-GFP and Opi1-mCherry puncta that colocalize. Bar = 4 μm. **(C and D)** Quantification of cells shown in B showing percent cells with Pex30-GFP and Opi1-mCherry accumulation in the indicated strains. Bars show mean from three independent experiments and SEM. 100 cells per genotype from each replicate were analyzed and compared using one-way ANOVA and Dunnett's multiple comparison test (****$P < 0.0001$). **(E)** 10-fold serial dilutions of *sei1pex30Δ* cells expressing Pex30-GFP truncation plasmids indicated in A were spotted on synthetic media without leucine. Cells were incubated for 2 days at 30°C and 37°C. **(F)** Western blot analysis of cell lysates from *sei1pex30Δ* cells expressing Pex30-GFP truncation plasmids indicated in A. Anti-GFP monoclonal antibody was used to detect Pex30 protein levels, and anti-Porin1 monoclonal antibody was used to detect porin levels as a control. **(G)** Quantification of protein levels from F. Bars show the mean from three replicates and SEM. One-way ANOVA and Dunnett's multiple comparison test were used to compare protein levels (*$P < 0.05$). Source data are available for this figure: SourceData F3. WF, widefield images.

studies that show PI3P accumulation at sites of LD biogenesis (Lukmantara et al., 2022).

Next, we confirmed the binding of Pex30 DysF domain to PA by performing a liposome flotation assay (Chowdhury et al., 2018; Melia et al., 2002). We incubated the purified 6xHis-DysF with liposomes composed of 100% DOPC or 90% DOPC and 10% PA and overlaid it with density gradients to determine

if the DysF domain floats with liposomes in PA-dependent manner. Upon ultracentrifugation, protein bound to liposomes will float to the top fractions of the gradient (Fig. 4 A). We used a known PA-binding C2 domain as a positive control. We found that DysF domain floats with DOPC + PA containing liposomes but not with liposomes containing only DOPC, suggesting DysF domain binds specifically to PA (Fig. 4 B). The liposomes

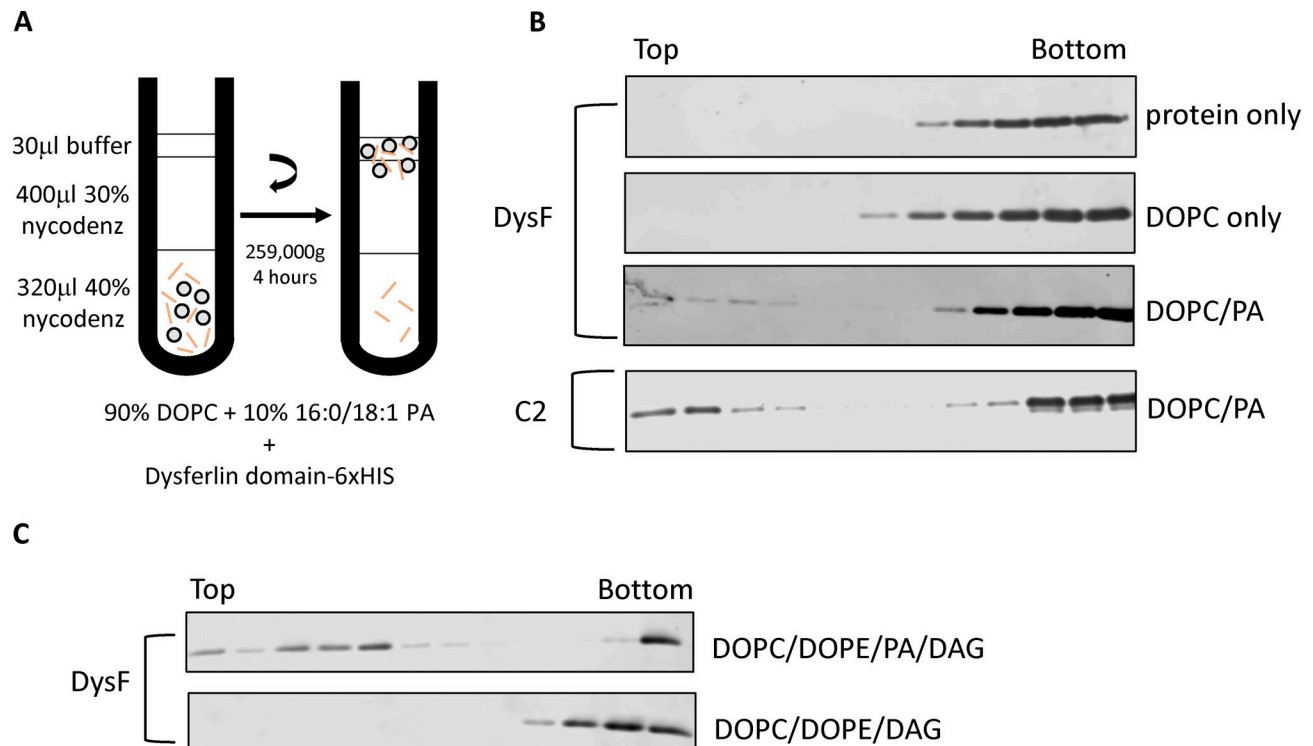

**Figure 4. DysF domain binds PA. (A)** Diagram of the liposome flotation assay based on density (Nycodenz) gradient centrifugation. **(B and C)** 12 fractions of 62.5 µl each were collected and subjected to western blot using anti-6xHis monoclonal antibody to check for the presence of 6xHis-DysF. Source data are available for this figure: SourceData F4. PE, phosphatidylethanolamine.

contained monounsaturated (PA 16:0, 18:1), whereas the lipid strip contains saturated PA (Fig. S2 D). Considering the binding of DysF domain to PA is weak on the lipid strip we checked if DysF binds to saturated PA using liposome flotation assay. We find that DysF also bound to PA (16:0, 16:0) and PA (16:0, 18:0) (Fig. S2 F), suggesting there is no preference to saturation as well as fatty acyl tail length of PA.

Previously, we demonstrated that Pex30 puncta colocalized with DAG sensor upon LD induction with oleic acid (Joshi et al., 2018). Therefore, we tested if DysF domain also binds DAG. We find that DysF domain bound to liposome containing DAG and PA but not to liposomes containing only DAG (Fig. 4 C). Purified perilipin 3 (provided by Dr. Michael Airola, Stony Brook University, New York, NY, USA), a known DAG-binding protein, was used as a positive control (data not shown) (Choi et al., 2023; Stribny and Schneiter, 2023). Together, our results demonstrate that Pex30 DysF domain specifically binds PA and not DAG (Fig. 4 C) suggesting PA is the primary driver of Pex30 recruitment at the ER subdomains. This supports our previous finding that Pex30 is localized at LD biogenesis sites (Joshi et al., 2018). As seipin also binds PA (Yan et al., 2018), it is possible that both Pex30 and seipin regulate the level and distribution of PA in the ER membrane.

### Molecular dynamics simulations demonstrate interaction between membrane and Pex30 DysF domain
Next, we conducted all-atom Molecular dynamics (MD) simulations to characterize interactions between the DysF domain and lipid bilayers with and without PA. We considered a pure DOPC bilayer and a mixed bilayer with 70% DOPC and 30% DOPA. The DysF domain was anchored to the lipid bilayer by two regions (Fig. 5, A–C; and Video 1). The region with the deepest insertion into the bilayer and the most lipid contacts consisted of a sequence of four hydrophobic residues (299–302) flanked by tryptophan residues (298 and 303), which have propensity to reside near the lipid-water interface. The other residues in this region (residues 296–315) tended to reside closer to the bilayer surface and have less sustained lipid contact. The second region (residues 379–397) maintained the most contact around a hydrophobic phenylalanine residue, flanked by polar and charged residues (390–395). The membrane-anchoring regions were the same for both membrane compositions, but there was an increase in the total number of lipid contacts with the protein when the bilayer contained PA (Fig. S3, A and B). This was particularly evident for two positively charged arginine residues (296 and 297) in the first region. The second region also had elevated lipid contact in the PC:PA bilayer. For the bilayer containing PA, we further characterized the propensity of each residue to bind PA versus PC by determining the fraction of total lipid contacts with PA (Fig. S3, C–E). Residues with contact percentage exceeding 30% exhibit more binding with PA than expected based on randomly distributed lipids. The contact of residues with the phosphate group of lipids, which is indicative of interactions with head groups, shows the most pronounced enhancement of PA interactions with DysF domain (Fig. S3 D).

**A**

β1

281 DSK**PIRFTYVLYENQRR**WLGIGWKPSMLSYERTPWTDEFLNEAPSPENFHLPEETNTMVWRWVDKTW
RLDMTNDGAIQVPNSKARTSADPSPDEGFI<u>YYDNTWKKPSKEDSFSK</u>**YTRRRRWVRTAELV**KT 410

β6

**B**

**C** Average total lipid contact

0 5

**D**

**E**

**F**

**G**

**H**

Figure 5. **All-atom MD simulations to characterize DysF domain and its membrane interactions. (A)** Amino acid sequence of the DysF domain, with residues involved in membrane anchoring underlined, and β1 and β6 strands depicted by bolded text. **(B)** Snapshot from a simulation of the DysF domain with a 70:30 DOPC:DOPA bilayer. The tails of DOPC and DOPA are shown in grey and blue, respectively. **(C)** Surface and cartoon representation of the DysF domain. Each residue is colored based on its average number of lipid contacts, with a contact defined as a lipid within 4.5 Å of the residue. **(D)** WF images of *sei1pex30Δ*

cells endogenously expressing Opi1-mCherry and GFP-tagged Pex30 (296–315Δ) or Pex30 (378–398Δ) plasmids grown to logarithmic phase. Bar = 4 µm. **(E)** Quantification of cells shown in D showing percent cells with Pex30-GFP accumulation in the indicated strains. Bars show mean from three independent experiments and SEM. 100 cells per genotype from each replicate were analyzed and compared using one-way ANOVA and Dunnett's multiple comparison test (***$P < 0.001$). **(F)** 10-fold serial dilutions of *sei1pex30Δ* cells expressing Pex30-GFP truncation plasmids were spotted on synthetic media without leucine. Cells were incubated for 2 days at 30°C and 37°C. **(G)** Western blot analysis of cell lysates from *sei1pex30Δ* cells expressing Pex30-GFP and DysF truncation plasmids in D. Anti-GFP monoclonal antibody was used to detect Pex30 protein levels, and anti-Porin1 monoclonal antibody was used to detect porin levels as a control. **(H)** Quantification of protein levels from G. Bars show the mean from two replicates and SEM. One-way ANOVA and Dunnett's multiple comparison test were used to compare protein levels. Source data are available for this figure: SourceData F5. WF, widefield images.

---

Based on MD simulations prediction, we deleted the two regions that tend to interact with membrane lipids to generate GFP-tagged Pex30 (296–315Δ) and Pex30 (378–398Δ). Both the truncations failed to accumulate at the ER-LD contact sites with Opi1-mCherry, indicating that these regions of DysF domain interact with PA (Fig. 5, D and E). Additionally, these truncations do not rescue the growth defect of *sei1pex30Δ* mutant, suggesting these residues are essential for Pex30 function (Fig. 5 F). Loss of function was not due to decreased protein levels, as Pex30 (296–315Δ)-GFP and Pex30 (378–398Δ)-GFP expression were comparable with Pex30-GFP (Fig. 5, G and H). Taken together, these results indicate that PA recruits Pex30 at the ER subdomain by interacting with two regions of the DysF domain.

### Pex30 and PA distribution is affected by PC levels in *sei1Δ* cells

We performed a targeted screen to test if other membrane phospholipids regulate Pex30 distribution. Here, we deleted genes from major phospholipid biosynthesis pathways to deplete specific phospholipids in *sei1Δ*-expressing Pex30-2xmCherry (Fig. 6 A) (Henry et al., 2012). Like in *sei1Δ*, Pex30-2xmCherry is accumulated as supersized puncta in *sei1ino1Δ* (Fig. 6, B and C). As *INO1* is involved in inositol synthesis (Culbertson et al., 1976), which is the headgroup of PI, our results indicate PI levels might not regulate Pex30 distribution in the cell. While deletion of *CHO2* in *sei1Δ* reduces Pex30-2xmCherry accumulation, *sei1opi3Δ* exhibits a significant restoration of Pex30-2xmCherry to WT-like (Fig. 6, B and C). This was surprising as Cho2 is upstream of Opi3 in the PC synthesis pathway (Fig. 6 A) (Henry et al., 2012). However, it has been reported that Opi3 can substitute Cho2 by methylating phosphatidylethanolamine to form phosphatidylmonomethylethanolamine in its absence, which is reflected in the PC levels as *cho2Δ* mutant has more PC than *opi3Δ* (Fig. S4 A) (Greenberg et al., 1983; Summers et al., 1988). To further confirm that PC levels affect the distribution of Pex30-2xmCherry, we grew *cho2Δ*, *sei1cho2Δ*, *opi3Δ*, and *sei1opi3Δ* in media supplemented with choline (Fig. 6 B) or in YPD (data not shown) to allow cells to utilize the Kennedy pathway for PC synthesis. While percent cells with Pex30-2xmCherry accumulation did not significantly increase in *sei1cho2Δ* and *sei1opi3Δ* when grown in media with choline supplementation (Fig. 6, B and C), a significant increase in cells with Pex30-2xmCherry accumulation in *sei1opi3Δ* when grown in YPD media was observed (data not shown). Our data suggest that acute choline supplementation might not be enough to restore Pex30-2xmCherry accumulation. It is possible that the Kennedy pathway is not efficient in *sei1Δ*, as enzymes such as Pct1 are enriched on to LDs (Grippa et al., 2015). Additionally, we find a significant increase in percent *sei1opi3Δ* cells with Pex30-2xmCherry

accumulation when *OPI3* was expressed, suggesting the Pex30-2xmCherry distribution in *sei1opi3Δ* was due to deletion of *OPI3* (Fig. 6 D). We confirmed the decrease in Pex30-2xmCherry accumulation in *sei1cho2Δ* is not because of phosphatidylethanolamine accumulation, a non-bilayer–forming phospholipid, by imaging Pex30-2xmCherry distribution in *psd1psd2sei1Δ*. The percent cells with Pex30-2xmCherry accumulation decrease in *psd1psd2sei1Δ*, which was restored by ethanolamine supplementation (Fig. S4, B and C). Next, we checked if the source of ectopic PA accumulation in *sei1Δ* is PC. Spo14, a phospholipase D, catalyzes hydrolysis of PC to form PA and choline (Sreenivas et al., 1998; Rose et al., 1995). We find that Pex30-2xmCherry and Opi1-GFP accumulated in *sei1spo14Δ* as in *sei1Δ*, suggesting that ectopic PA accumulation in *sei1Δ* is not due to PC hydrolysis by Spo14 (Fig. S4, D and E). The source of ectopic PA accumulation in the *sei1Δ* mutant remains unknown.

As Pex30-2xmCherry distribution was restored to WT-like in *sei1opi3Δ*, we determined if ectopic PA accumulation in *sei1opi3Δ* cells is also altered. We expressed Opi1-GFP in the presence and absence of choline in *sei1opi3Δ*. In the absence of choline, *sei1opi3Δ* exhibited fewer Opi1-GFP puncta as compared with *sei1Δ* (Fig. 6, E and F). Interestingly, the total cellular PA levels in *sei1opi3Δ* cells are higher than *sei1Δ*, suggesting that a decrease in PC levels affects the ectopic accumulation as well as cellular levels of PA in *sei1opi3Δ* cells (Figs. 6, E and F; and S4, A and F). Upon addition of choline, percent cells with Opi1-GFP puncta formation increases in *sei1opi3Δ*, but not to the extent of *sei1Δ* (Fig. 6, E and F). Together, these results show that cellular PC levels in the *sei1Δ* mutant affect PA and Pex30-2xmCherry accumulation at ER-LD contact sites.

### Discussion

In this study, we demonstrate how Pex30, an ER membrane–shaping protein, is recruited at the specialized ER subdomains to drive LD biogenesis. Using the *sei1Δ* mutant, which exhibits ectopic PA accumulation at ER-LD contact sites, we demonstrate that PA recruits Pex30 to these sites by binding to the DysF domain (Fig. 7 A). Using in vitro liposome flotation assay, we show DysF domain binds to various PA species. Furthermore, MD simulations demonstrated that DysF domain stably inserts in the membrane using two regions containing hydrophobic amino acid residues, while the adjacent positively charged arginine residues preferentially bind to PA. Unlike seipin, Pex30 specifically binds PA but not DAG. Both seipin and Pex30 also regulate cellular PA levels, as loss of both leads to accumulation of PA, especially species with monounsaturated fatty acids. We propose that PA regulates the spatiotemporal

Figure 6. **Pex30 and PA distribution is affected by PC levels in *sei1Δ* cells. (A)** Yeast phospholipid synthesis pathway. The genes targeted in the screen are indicated in red. **(B)** WF images of the indicated strains endogenously expressing Pex30-2xmCherry. Cells were grown in synthetic media with or without 2 mM choline supplementation until logarithmic phase. Yellow arrowheads denote Pex30-2xmCherry accumulation. Bar = 4 µm. **(C)** Quantification of experiment in B showing percent cells with Pex30-2xmCherry accumulation. Bars show mean from three independent experiments and SEM. 100 cells from each replicate were analyzed and compared using two-way ANOVA and Tukey's multiple comparison test. Means were compared against *sei1Δ*. Additional comparisons with choline treatment are shown and are not significant (*P < 0.05, ****P < 0.001). **(D)** WF images of *sei1opi3Δ* cells in logarithmic phase endogenously expressing Pex30-2xmCherry and *OPI3* on a plasmid. Yellow arrows denote Pex30-2xmCherry accumulation. The graph on the right is quantification of Pex30-2xmCherry accumulation in *sei1opi3Δ* with or without overexpression of *OPI3*. Bars show mean from two independent experiments. 100 cells from each replicate were analyzed. Bar = 4 µm. **(E)** WF images of the indicated strains expressing Opi1-GFP on a plasmid. Cells were grown in synthetic media with or without 2 mM

choline supplementation. White arrowheads denote Opi1-GFP accumulation. Bar = 4 µm. **(F)** Quantification of experiment in E showing percent cells with Opi1-GFP puncta. Bars show mean from three independent experiments and SEM. 100 cells per genotype from each replicate were analyzed and compared using two-way ANOVA and Tukey's multiple comparison test (*$p < 0.05$ and **$p < 0.01$). WF, widefield images.

distribution of Pex30 at ER subdomains that drive LD biogenesis (Fig. 7 B).

Formation of new LDs involves assembly of multiple factors, including proteins such as TAG synthases, seipin, Nem1, Erg6, Pex30, and Pet10 and membrane curvature–inducing lipids. It was reported that seipin and Nem1 establish the ER subdomains to initiate LD formation and sequential recruitment of other factors to these sites (Choudhary and Schneiter, 2020). Our findings demonstrate that new LDs form at Pex30 punctae independent of seipin (Fig. 2, A–C). What drives LD formation in the absence of seipin? Our previous findings show that Pex30 is an ER membrane–shaping protein localized to ER subdomains where new LDs form (Joshi et al., 2016; Joshi et al., 2018). Here, we demonstrate that Pex30 accumulates at ER subdomains associated with LDs in the *sei1Δ* mutant (Fig. 7 A). These subdomains are also sites of ectopic accumulation of PA. We propose Pex30 and PA drive LD biogenesis in the absence of seipin. What is the role of Pex30 at ER subdomains? Considering Pex30 has a RHD, it is possible that Pex30 could locally induce membrane curvature to enrich PA and PA-synthesizing enzymes at the ER-LD contact sites and regulate cellular PA levels. It is also possible that Pex30 lowers the surface tension at ER subdomains enriched with PA accumulation to favor LD formation. In the absence of Pex30, cells form several small and highly clustered LDs, suggesting Pex30 is required for LD biogenesis. *PEX30* also exhibits negative genetic interaction with *SEI1*, suggesting the function of Pex30 is important in *sei1Δ* (Joshi et al., 2018). How is Pex30 recruited to the ER subdomains? We show that Pex30 is recruited by PA to the ER subdomains by binding to the Pex30 DysF domain. There is sequence similarity between yeast and human DysF domains, especially at the arginine/tryptophan repeats (Sula et al., 2014). Patients with mutations in DysF protein exhibit accumulation of LDs, suggesting human DysF protein affects lipid metabolism (Li et al., 2016; Grounds et al., 2014). In fact, mutations in human DysF protein linked to dysferlinopathies are mostly in the DysF domain. Whether human DysF domain as well as DysF domains from other Pex30-like proteins also binds PA remains to be determined. We also find weak binding of DysF to PI3P, PI4P, and PI4,5P2 (Fig. S2 D), suggesting this domain could regulate localization of Pex30 at multiple MCSs through lipid binding (Ferreira and Carvalho, 2021).

Based on our findings, we propose that Pex30 distribution in the ER membrane at LD biogenesis sites is regulated by PA (Fig. 7 B). Even though we demonstrate how Pex30 accumulates at PA-enriched sites in the *sei1Δ* mutant, we believe our findings could be extrapolated to WT cells (Fig. 6, A and B). The Pex30 RHD domain could generate membrane curvature, which is essential for PA enrichment and LD formation. Considering there is delay in LD formation as well as recruitment of TAG synthases in the *pex30Δ* mutant, the membrane curvature generated by Pex30 could be required for recruitment of LD biogenesis enzymes and lipids. Future studies should focus on characterizing the Pex30 ER subdomains in different growth conditions and mutant backgrounds. Like Pex30, the distribution of its functional mammalian homologs MCTP1 and MCTP2 at multiple MCSs could also be regulated by phospholipids (Joshi et al., 2021). Thus, PA determines the distribution of the LD biogenesis sites in the ER membrane, whereas Pex30 and seipin maintain phospholipid homeostasis and surface tension at the discrete ER subdomains.

## Materials and methods

### Yeast strains and plasmids

Yeast strains, plasmids, and primers used in this study are listed in Tables S1, S2, and S3, respectively. Gene knockouts were generated using PCR-based targeted homologous recombination and tetrad dissection techniques. Knockouts using PCR-based homologous recombination were generated by replacing the ORF of the gene with a selection cassette amplified from PCR (Longtine et al., 1998). Knockouts by tetrad dissection were generated by crossing haploid cells of opposite mating types, inducing sporulation of the cells, dissecting tetrads, and confirming genotype using PCR. The *psd1psd2Δ* strain and FYVE-GFP

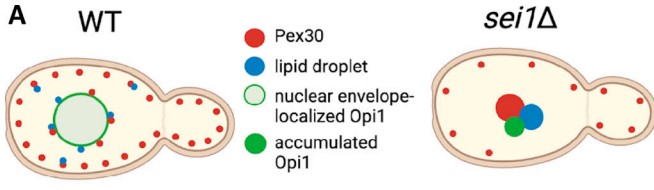

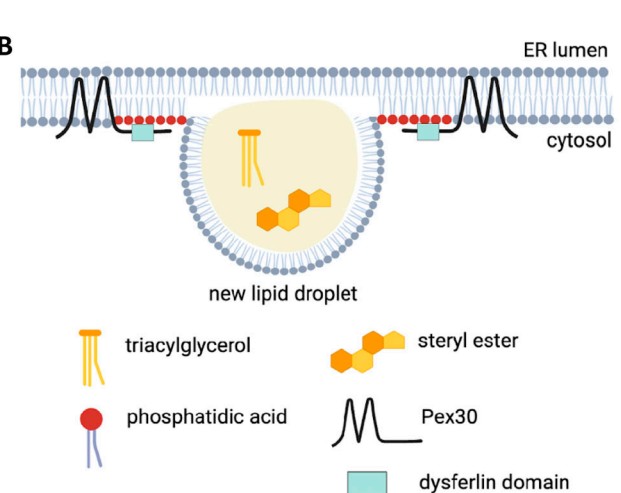

Figure 7. **Proposed model for recruitment of Pex30 at LD biogenesis sites in the ER subdomains. (A)** Distribution of Pex30, Opi1-GFP (PA sensor), and LDs in WT and *sei1Δ*. **(B)** PA recruits Pex30 at the ER-LD contact sites by binding to the DysF domain.

plasmid was a gift from Will Prinz lab (UT Southwestern medical center, Dallas, TX, USA). Strains with endogenously expressed fluorescent proteins were also generated using PCR-based homologous recombination. Knockout cassettes and tagging cassettes were transformed into yeast cells using the lithium acetate method of transformation. Plasmids generated in this study were constructed by double restriction enzyme digestion of the plasmid at BamHI and SalI restriction sites. DNA fragments of interest were amplified using PCR with primers, including overhangs homologous, to the restriction sites. Digested plasmids and PCR products were transformed into yeast cells using the lithium acetate method of transformation, and ligated plasmids were isolated and amplified in competent *Escherichia coli* cells (C3030H; New England Biolabs). Whole-plasmid sequencing was performed at Plasmidsaurus.

## Yeast media and growth conditions
Yeast cells were grown in YPD media (1% yeast extract, 2% peptone, and 2% glucose) or synthetic complete (SC) media (0.67% yeast nitrogen base without amino acids [USBiological], amino acid mix [USBiological], and 2% glucose). In LD induction experiments, SC media was prepared using 2% raffinose or 2% galactose. Choline (Acros Organics) or ethanolamine (Sigma-Aldrich) was added to SC media at 2 or 1 mM concentration, respectively, when indicated. *sei1pex30Δ* cells were cultured at 25°C, unless stated otherwise, and all other strains were cultured at 30°C.

## LD induction experiment
Cells were precultured in SC media containing 2% raffinose to stationary phase and washed twice with sterile MilliQ water before switching cells to SC media containing 2% galactose for LD induction. Cells were imaged before galactose addition and after galactose addition every hour for 6 h and subsequently after 14 h. MDH dye (Abcepta) was added to cultures 1 h before imaging at final concentration of 0.1 mM, and cells were pelleted and washed twice with PBS before imaging.

## Fluorescence microscopy
For fluorescence microscopy, cells were grown in overnight cultures, diluted to 0.2–0.3 $OD_{600}$ units, and grown until mid-logarithmic phase for imaging, unless indicated otherwise. For LD visualization, MDH dye was added to the media at a concentration of 0.1 mM, cells were incubated for 30 min at 30°C, pelleted, washed twice with 1X PBS, and pipetted on glass slides for live-cell imaging. Images were acquired on an inverted Zeiss 900/Airyscan laser scanning confocal microscope equipped with Colibri 7-channel solid-state fluorescence light source with two filter sets for widefield microscopy and diode lasers and gallium arsenide phosphide and Airyscan detectors for Airyscan confocal microscopy. Images were acquired using a 63×/1.4 NA objective lens. Airyscan images were processed by Airyscan processing using the Zeiss ZEN software package.

## Image quantification
All images were analyzed using ImageJ software. Quantification of images was conducted manually of Z-stacked images. All

statistics were performed in GraphPad Prism 9.5.1. Percent values were generated by dividing the number of cells showing a specific phenotype by the total number of cells analyzed. The specific statistical analyses conducted on each dataset are indicated in figure legends.

## DysF domain expression and purification
The DysF domain of Pex30 (280–410 amino acid) was cloned into the pET15b vector and expressed in the *E. coli* T7 expression PLYSS strain. A single colony of *E. coli* was cultured overnight in Luria–Bertani medium with ampicillin (100 µg/ml) and chloramphenicol (25 µg/ml) at 37°C. From the preculture, 5% inoculum was added to the fresh Luria–Bertani medium with the same antibiotic concentration and incubated until the $OD_{600}$ reached 0.6. The protein expression was then induced by the addition of 0.4 mM IPTG and further incubated at 30°C for 3 h. Cells were harvested by centrifugation at 10,000 rpm for 10 min at 4°C, the pellet was thawed, and cells were resuspended in PBS and lysed using a French press at 25,000 psi. The cleared lysate was subjected to Ni-NTA (Thermo Fisher Scientific) affinity chromatography, including washing at least 10 times with wash buffer (25 mM imidazole and 1X PBS buffer) and eluted with elution buffer (500 mM imidazole and PBS buffer). The Ni-NTA purified protein was further purified using size exclusion chromatography.

## Liposome preparation and liposome flotation assay
Lipids used in this study were purchased from Avanti Polar Lipids. Lipids were mixed in a clean glass tube, dried from chloroform stock solutions, and dried under gentle argon stream. The thin lipid film obtained was further dried overnight under a vacuum and then hydrated in 25 mM HEPES (pH 7.3) and 150 mM NaCl, followed by four freeze–thaw cycles using liquid nitrogen and at 42°C water bath. The dissolved solution was further passed >30 times through a 100-nm polycarbonate filter membrane using the extruder from Avanti Polar Lipids, Inc. For DAG containing liposomes in Fig. 4 C, the dried lipid film was resuspended in liposome buffer (50 nM NaCl and 25 mM Tris, pH 7.5), resulting in a concentration of 2 mM phospholipids (70% DOPC, 20% DOPE, 10% DAG and 60% DOPC, 20% DOPE, 10% PA, 10% DAG). The phospholipid suspension was then subjected to 10 cycles of freezing in liquid nitrogen and thawing in a water bath at 42°C. The resulting multilamellar liposomes were extruded 30 times through a polycarbonate filter of 0.1-µm pore size to generate unilamellar vesicles.

A liposome flotation assay was done as described previously (Maeda et al., 2019). Briefly, 20 µl DysF domain at a final concentration of 2.5 µM was mixed with 140 µl of liposome (1 mM) in 25 mM HEPES (pH 7.3), 150 mM NaCl, and 0.5 mM DTT and incubated at 4°C for 1 h. After incubation, an equal volume of 80% Nycodenz (Axell) was added to 160 µl of protein–liposome mixture to make it 320 µl of 40% Nycodenz solution. A layer of 400 µl of 30% Nycodenz was placed on the top of the bottom layer, and 30 µl of buffer with no Nycodenz solution was placed on the top (25 mM HEPES, pH 7.3, 150 mM NaCl, and 0.5 mM DTT). The tubes were centrifuged in SW 55 Ti rotor (Beckman Coulter) at 55,000 RPM for 4 h at 4°C. For Fig. 4 C, the assay was

performed as described previously (Stribny and Schneiter, 2023). Briefly, the purified DysF at 4 µM was incubated with liposomes 2 mM for 1 h at room temperature in 60 µl volume and then gently mixed with an equal volume of 60% (wt/vol) sucrose solution in liposome buffer to obtain a final sucrose concentration of 30%. This mixture was overlaid with 2 vol of 20% sucrose solution, 2 vol of 10% sucrose solution, and 1 vol of liposome buffer. The samples were centrifuged at 180,975 g for 1 h at 20°C. After centrifugation, 12 fractions of 62.5 µl were collected and subjected to western blot analysis using anti-6XHis monoclonal antibody (Invitrogen). The blots were scanned on a Li-COR Odyssey scanner (Li-COR biosciences).

### Protein lipid overlay assay

For Fig. S2 D, the PIP lipid strips were purchased from Echelon Biosciences. The lipid binding assay was performed according to the manufacturer's protocol with slight modification. Briefly, 1 µl of purified DysF domain was spotted at the bottom corner of the strip as a control and left in the dark until completely dried. The lipid strip was then blocked using a blocking buffer (PBS-0.1% Tween20 + 3% BSA) for 1 h at room temperature with gentle agitation. The buffer was discarded, and purified DysF domain at a final concentration of 2.5 µg/ml in the blocking buffer was added and left for overnight incubation at 4°C with gentle agitation. The protein solution was then discarded, and the strip was washed three times with a wash buffer (PBS-0.1% Tween20) for 10 min each with gentle agitation. The wash step is followed by the addition of an anti-mouse monoclonal antibody for 6XHis tag at a 1:2,000 dilution in the blocking buffer and incubated overnight at 4°C with gentle agitation. The strip was washed three times with a wash buffer, followed by the addition of a donkey anti-mouse antibody at a dilution of 1:5,000 in the blocking buffer for 1 h at room temperature with gentle agitation. After washing three times with a wash buffer, the protein was detected by using a Li-COR Odyssey scanner (Li-COR biosciences).

### Protein extraction and western blot analysis

For Figs. 3, F and G; 5, G and H; and S1, A and B, a total of 1 OD unit of yeast cells from each strain were pelleted. Pellets were resuspended in a 2 M LiAc solution and incubated on ice for 5 min. Cells were pelleted and resuspended in a 0.4 M NaOH solution and incubated on ice for 5 min. Cells were pelleted, resuspended in SDS sample buffer (Laemmli sample buffer [Bio-Rad] and βME), and boiled for 5 min. An equal volume of each sample was loaded on an SDS-PAGE gel, transferred to a 0.2-µm nitrocellulose membrane, and blocked for 1 h in blocking buffer (5% skim milk in TBST). The membranes were incubated with primary antibody (GFP [Roche] 1:5,000; porin [Invitrogen] 1:2,000) in blocking buffer overnight at 4°C, washed three times with TBST, and incubated with secondary antibody for 1 h. Following three washes with TBST, the membranes were imaged using a Li-COR Odyssey scanner (Li-COR biosciences). Western blots were analyzed and quantified using Image Studio analysis software. Each sample was normalized by dividing the intensity of the sample by the intensity of the respective porin. To compare mutant protein levels with WT, the mutant normalized value was divided by the WT normalized value.

### All-atom MD simulations

We considered two membrane compositions (pure DOPC and 70:30 DOPC: DOPA) and used the CHARMM-GUI membrane builder to set up lipid bilayers consisting of 720 lipids (Jo et al., 2008; Jo et al., 2009). The structure of the DysF domain was predicted using ColabFold on the full Pex30 sequence (Mirdita et al., 2022). The N terminus and C terminus of the DysF domain from the ColabFold prediction were patched with acetyl and methylamine groups, respectively, to avoid end effects in simulations. The resulting DysF domain was then inserted into the bilayer using CHARMM-GUI. The positioning of proteins in membranes server was used to determine the initial placement of the protein (Lomize et al., 2012). All simulations were performed in GROMACS 2018 using the CHARMM36 force field (Van Der Spoel et al., 2005; Abraham et al., 2015; Lee et al., 2016; Klauda et al., 2010). The temperature was set at 303.15 K to ensure all lipids were in the liquid-disordered state, and a concentration of 0.15 mM KCl was used to neutralize the charge of the system.

The energy of the system was first minimized using steepest descent for 5,000 steps. Subsequently, the system was equilibrated in six stages using the standard equilibration procedure from CHARMM-GUI. During this procedure, the simulation time step was increased from 1 to 2 fs while using the Berendsen thermostat and Berendsen barostat (Berendsen et al., 1984). Subsequently, we simulated the systems for 400 ns using the Nosé-Hoover thermostat and Parrinello–Rahman barostat (Nosé, 1984; Hoover, 1985; Parrinello and Rahman, 1981). The final 200 ns was used for analysis.

Trajectories generated from GROMACS were analyzed in Python using the MDAnalysis package (Michaud-Agrawal et al., 2011). To analyze contacts, all lipids within a cutoff distance were counted around each residue. The cutoff for contact was set at 4.5 Å. The percentages of each type of lipid in contact were then calculated for residues with an average total lipid contact >0.5 lipids.

### Phospholipid extraction

Lipid extraction was performed following the procedure detailed by Yang et al. (2022). Briefly indicated strains were grown to OD 1.0 in 10 ml SC media. Cells were centrifuged, washed once with sterile deionized water, flash frozen in liquid nitrogen, and stored at –80°C. Cell pellets were then resuspended in 500 µl of deionized water, transferred to tared vials, and lyophilized overnight. The following day, the pellets were measured and resuspended in 500 µl of ice-cold methanol and 10 µl of 5 ng/µl of EquiSPLASH Lipidomix (Avanti Polar Lipids) containing 5 ng/µl of 15:0/18:1-d7-phosphatidic acid (Avanti Polar Lipids) was added. The resuspensions were then transferred to microcentrifuge tubes containing 400 µl of glass beads and agitated in a CryoMill (Retsch) for 5 min at 30 Hz twice with a 5-min period of cooling on ice in-between agitations. The resulting cell lysate were filtered into a new microcentrifuge tube, and the remaining glass beads were washed with an additional 500 µl of ice-cold methanol. This wash was combined with the lysate and then transferred to a 15-ml centrifuge tube where 2.0 ml of chloroform was then added. The lysates were vortexed for 30 s

**Table 1. Chromatography gradient**

| Time (min) | % A | % B |
|---|---|---|
| 0.00 | 70.00 | 30.00 |
| 5.00 | 57.00 | 43.00 |
| 5.10 | 50.00 | 50.00 |
| 14.00 | 30.00 | 70.00 |
| 21.00 | 1.00 | 99.00 |
| 24.00 | 1.00 | 99.00 |
| 24.10 | 70.00 | 30.00 |
| 28.00 | 70.00 | 30.00 |

and then centrifuged at 1,000 $g$ for 3 min. The supernatants were transferred to fresh 15-ml centrifuge tubes, where 400 µl of 50 mM citric acid was added, followed by an additional 800 µl of chloroform. The solution was vortexed for 30 s and then centrifuged at 1,000 $g$ for 10 min to achieve phase separation. The bottom layer was removed and transferred to a 4-ml glass vial, and the solvent was evaporated under a stream of nitrogen. The residue was then resuspended in 100 µl of 2:1:1 isopropanol/acetonitrile/water and transferred to an autosampler vial for liquid chromatography high-resolution mass spectrometry analysis.

### Liquid chromatography high-resolution mass spectrometry analysis

Chromatography was conducted using a Vanquish UHPLC system equipped with a Thermo Accucore C30 column (150 mm × 2.1 mm, 2.6 µm). Separations were conducted using a 28-min gradient with a flow rate of 0.350 ml/min given the program below, in which mobile phase A is 60:40 acetonitrile/water with 10 mM ammonium formate and 0.1% formic acid, and mobile phase B is 90:10 isopropanol/acetonitrile with 10 mM ammonium formate and 0.1% formic acid (Table 1).

Mass spectra were acquired using a Thermo Scientific Exploris 120 mass spectrometer equipped with an electrospray ionization probe in negative ion mode given the following source parameters: spray voltage (V): 3,000, sheath gas (arb): 7, sweep gas (Arb): 1, ion transfer tube temp. (°C): 350, and vaporizer temp. (°C): 400.

Data dependent acquisition was done with an MS1 resolution of 120,000 and a scan range of 150–1,700 m/z. RF lens was set to 50%. Dynamic exclusion was set to exclude precursors when measured one time for a duration of 15 s and a mass tolerance window of ±5 ppm. MS2 spectra were acquired from the top 4 ions from the preceding parent scan at 30,000 resolution and an isolation window of 1.2 m/z. Fragmentation was produced using stepped, normalized collision energies of 20%, 24%, and 28%.

Mass spectrometry data were analyzed using MS-DIAL to match the acquired MS/MS spectra to reference spectra in the LipidBLAST database (Tsugawa et al., 2015; Kind et al., 2013). Annotated spectra of reference-matched phospholipids were manually curated. Quantitation was performed by normalizing peak areas to the added internal standards. Statistical significance was determined by performing one-way ANOVA.

## Online supplemental material

Fig. S1 is associated with Fig. 1, which shows that Pex30 and PA accumulate at ER-LD contact sites in *sei1Δ*. Fig. S2 is associated with Fig. 4, which shows that DysF domain binds PA. Fig S3 is associated with Fig. 5, which shows the all-atom MD simulation to characterize DysF domain binding to PA. Fig S4 is associated with Fig. 6, which shows that Pex30 and PA distribution is affected by PC levels in *sei1Δ* cells. Video 1 shows the all-atom simulation video of 200 ns of DysF domain binding to the membranes. Tables S1, S2, and S3 lists the yeast strains, plasmids, and primers used in this study.

## Acknowledgments

We thank Drs. William Prinz and Sarah Cohen for reading the manuscript. We thank the Biological and Small Molecule Mass Spectrometry Core (BSMMSC) at UTK for providing instrumentation for LC-HRMS/MS analysis.

Research reported in this publication was supported by the National Institutes of Health award R35 GM147189 and startup funds from the University of Tennessee at Knoxville to A.S. Joshi. M. House was supported by NIH T32 award GM142621. Computer simulations were performed on the University of Tennessee Infrastructure for Scientific Applications and Advanced Computing computational resources.

Author contributions: M. House: conceptualization, formal analysis, investigation, project administration, visualization, and writing—review and editing. K. Khadayat: investigation. T.N. Trybala: formal analysis, investigation, resources, and visualization. N. Nambiar: formal analysis, investigation, methodology, software, visualization, and writing—original draft. E. Jones: investigation. S.M. Abel: formal analysis, supervision, and writing—original draft, review, and editing. J. Baccile: investigation and resources. A.S. Joshi: conceptualization, formal analysis, funding acquisition, investigation, methodology, project administration, resources, supervision, validation, visualization, and writing—original draft, review, and editing.

Disclosures: The authors declare no competing interests exist.

Submitted: 28 May 2024

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

# Supplemental material

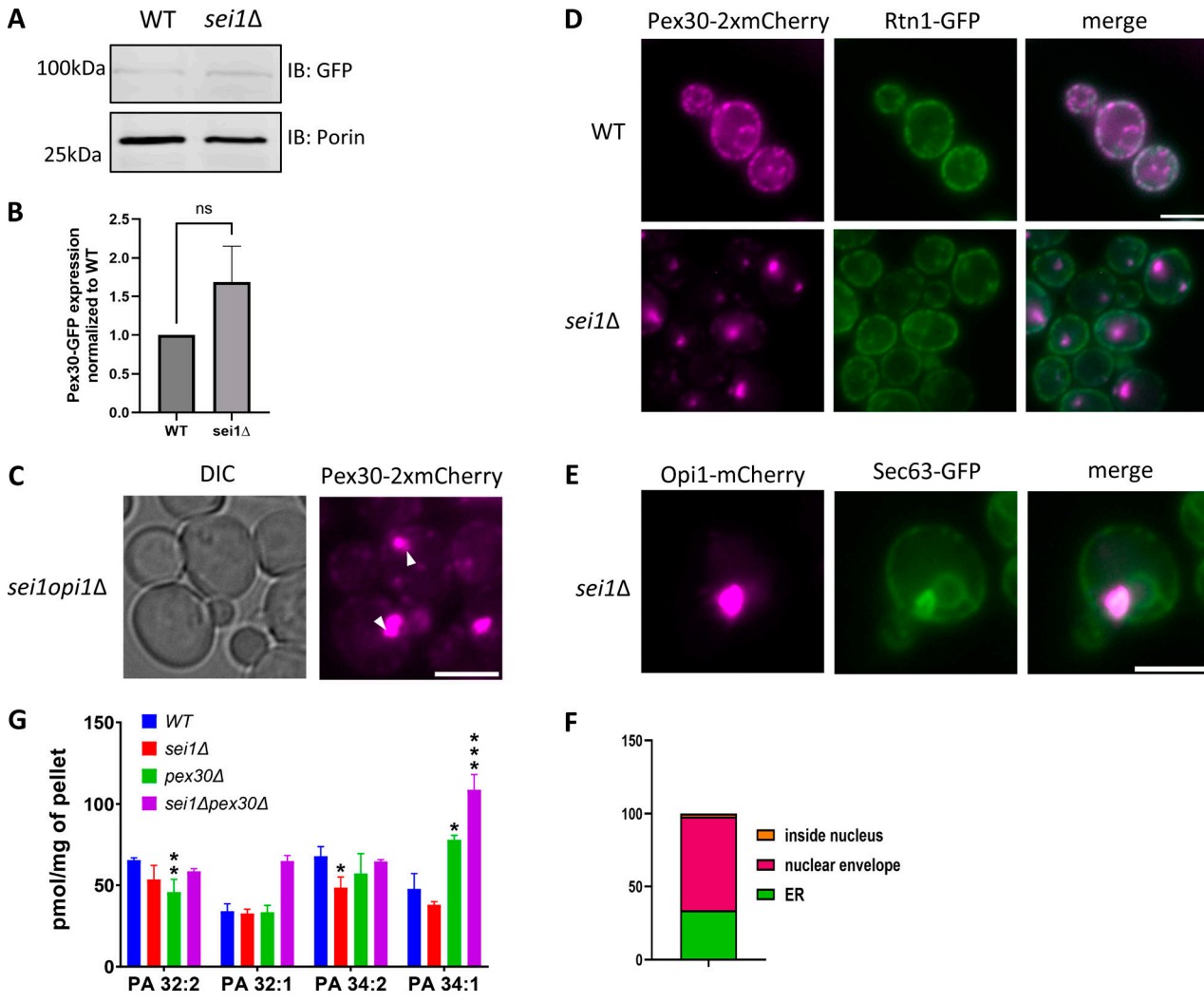

Figure S1. **Data associated with** Fig. 1. **(A)** Western blot analysis of cell lysates from WT and sei1Δ cells endogenously expressing Pex30-GFP. Anti-GFP monoclonal antibody was used to detect Pex30 protein levels, and anti-Porin1 monoclonal antibody was used to detect porin levels as a control. **(B)** Quantification of protein levels from A. Bars show the mean from three replicates and SEM. One-way ANOVA and Dunnett's multiple comparison test were used to compare protein levels. **(C)** WF images of sei1opi1Δ cells endogenously expressing Pex30-2xmCherry in logarithmic phase. White arrowheads show Pex30 accumulation. Bar = 4 µm. **(D)** WF images of WT and sei1Δ cells endogenously expressing Pex30-2xmCherry and Rtn1-GFP in logarithmic phase. Bar = 4 µm. **(E)** WF image of sei1Δ cells endogenously expressing Opi1-mCherry and Sec63-GFP on a plasmid as an ER marker in logarithmic phase. Bar = 4 µm. **(F)** Quantification of the localization of Opi1-mCherry puncta from E. **(G)** Phospholipid measurements of indicated strains (n = 3) by LC-HRMS. Distribution of total amounts of annotated PA species ($*P < 0.05$, $**P < 0.01$, and $***P < 0.001$). Source data are available for this figure: SourceData FS1.. WF, widefield images; LC-HRMS, liquid-chromatography high-resolution mass spectrometry

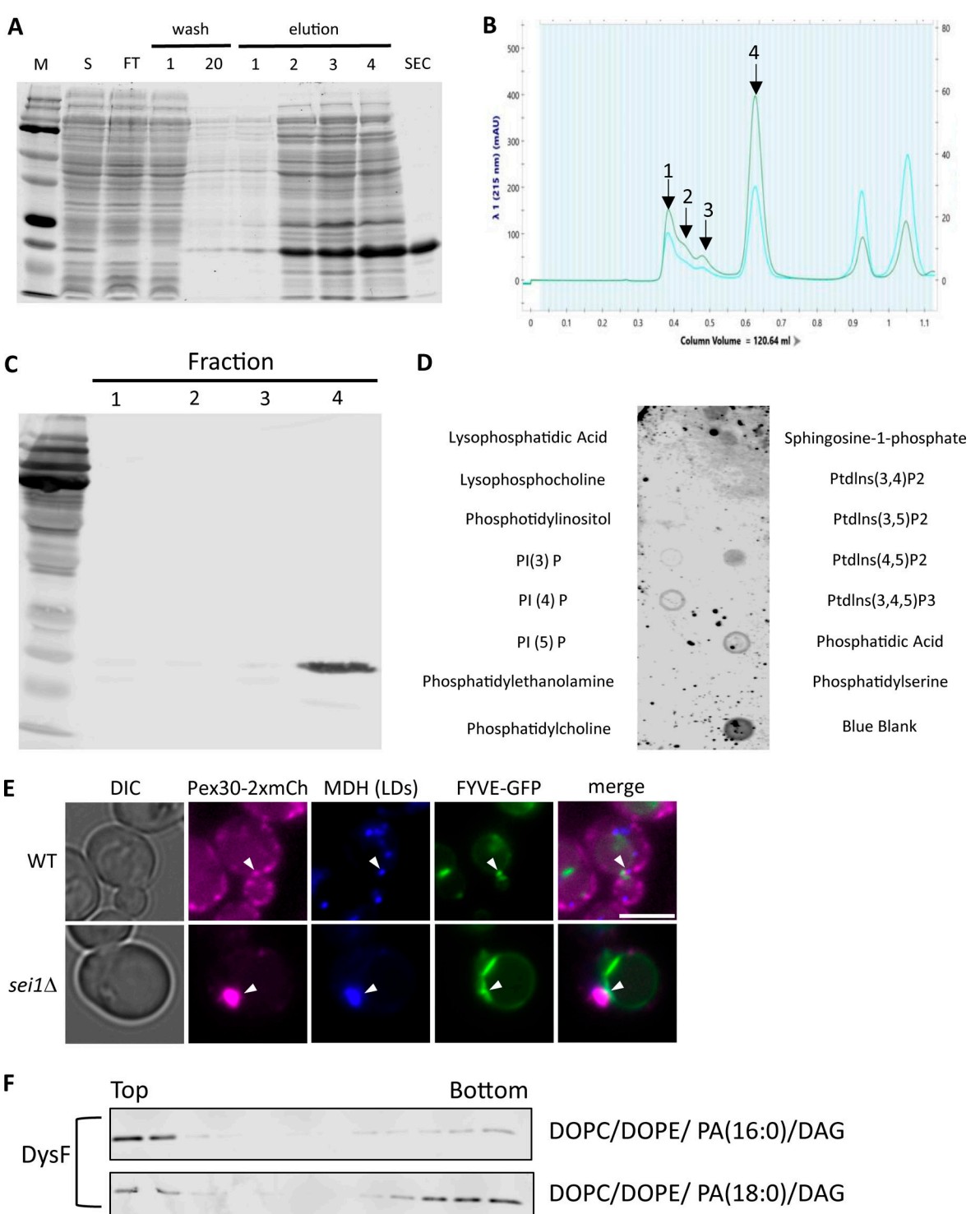

Figure S2. **Data associated with** Fig. 4. **(A)** SDS-PAGE gel stain with Coomassie blue simple stain. Protein ladder (M), lysate (S), flow through (FT), 1–20 wash, 1–4 elution with imidazole, and size exclusion chromatography (SEC). **(B)** Size exclusion chromatography. Peak 4 was collected and used for liposome flotation assays. **(C)** Protein fraction from B was immunoblotted using anti-6xHis antibody to check for the presence of 6xHis-DysF domain. **(D)** Protein-lipid overlay assay. **(E)** WF images of WT and sei1Δ cells endogenously expressing Pex30-2xmCherry and FYVE-GFP on a plasmid in logarithmic phase. Cells were stained with MDH for LDs. White arrowheads denote Pex30, LD, and FVYE puncta colocalization. Bar = 4 μm. **(F)** As in Fig. 4 C, but with different PA species. Source data are available for this figure: SourceData FS2.. WF, widefield images.

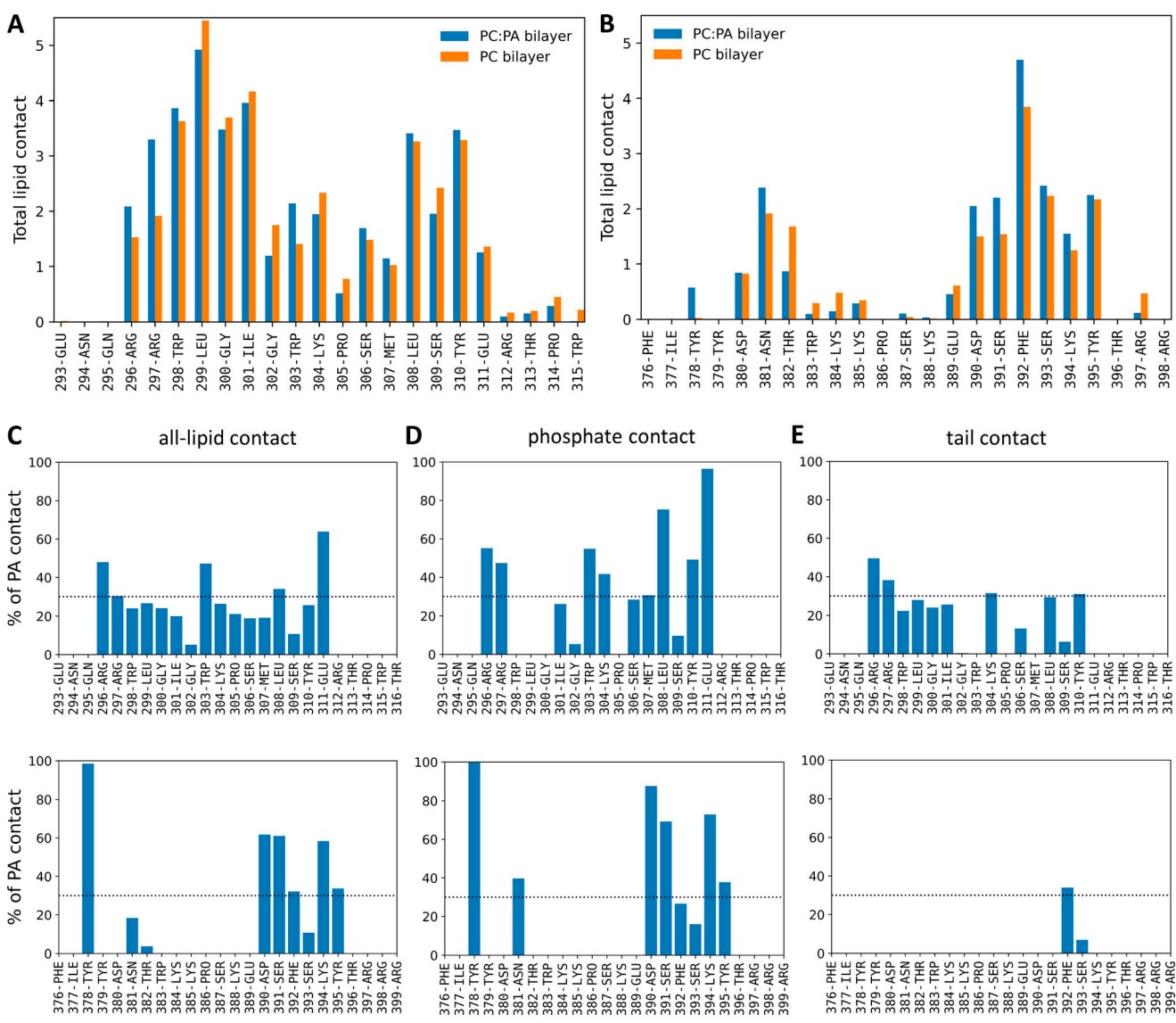

Figure S3. **Data associated with** Fig. 5. **(A and B)** Average number of lipids in contact with each residue of the DysF domain. Results for both pure DOPC and 70:30 DOPC:DOPA bilayers are shown. Only regions containing residues with lipid contacts are displayed. **(C–E)** Percentage of DOPA lipid contacts for each residue in the 70:30 DOPC:DOPA bilayer. Results are shown only for residues that have an average total lipid contact >0.5; no bar is shown otherwise. The horizontal dashed line denotes the overall composition of DOPA in the membrane. The panels distinguish between contacts with all atoms of the lipid (C), only atoms of the phosphate group (D), and only atoms in the tail (E). All contact analysis was conducted over 200 ns of simulation.

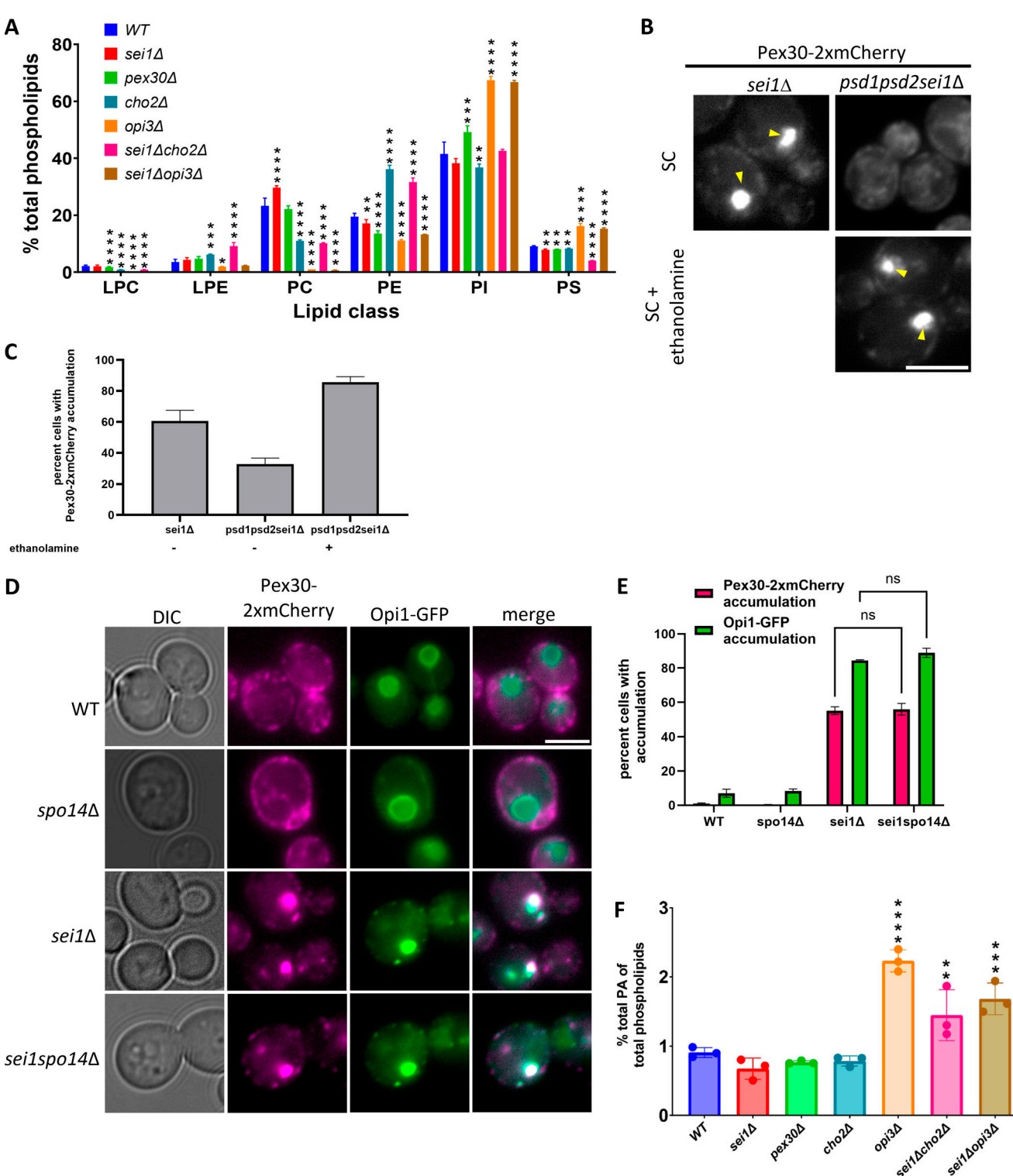

Figure S4.    **Data associated with** Fig. 6. **(A)** Phospholipid measurements of indicated strains taken by LC-HRMS (*n* = 3). Distribution of total quantitated phospholipids by class (*P < 0.05, **P < 0.01, ***P < 0.001, and ****P < 0.0001). **(B)** WF images of the indicated strains endogenously expressing Pex30-2xmCherry. Cells were grown in synthetic media or synthetic media with 1 mM ethanolamine supplementation. Yellow arrowheads denote Pex30 accumulation. Bar = 4 µm. **(C)** Quantification of experiment from B showing percent cells with Pex30 accumulation. Bars show mean from two independent experiments and SEM. 100 cells per genotype from each replicate were analyzed. **(D)** WF images of the indicated strains endogenously expressing Pex30-2xmCherry and Opi1-GFP on a plasmid in logarithmic phase. Bar = 4 µm. **(E)** Quantification of experiment from D showing percent cells with Pex30-2xmCherry accumulation and Opi1-GFP accumulation in each genotype. Bars show mean from three independent experiments and SEM. 100 cells per genotype from each replicate were analyzed and compared using one-way ANOVA and Tukey's multiple comparison test. **(F)** Phospholipid measurements of indicated strains taken by LC-HRMS (*n* = 3). Amount of total quantitated PA relative to total quantitated phospholipids (**P < 0.01, ***P < 0.001, and ****P < 0.0001).. WF, widefield images; LC-HRMS, liquid-chromatography high-resolution mass spectrometry.

Video 1.   **Data associated with** Fig. 5**.** All-atom simulation video of 200 ns of DysF domain binding to the membranes.

**Provided online are Tables S1, S2, and S3. Table S1 shows the yeast strain list. Table S2 shows the plasmid list. Table S3 shows the primer list.**

