## [Peer Review File · The Journal of Cell Biology]

Phosphatidic acid drives spatiotemporal distribution of Pex30 at ER-LD contact sites

Morgan House, Karan Khadayat, Thomas Trybala, Nikhil Nambiar, Elizabeth Jones, Steven Abel, Joshua Baccile, and Amit Joshi

Corresponding Author(s): Amit Joshi, University of Tennessee at Knoxville

Review Timeline:

Submission Date:	2024-05-28
Editorial Decision:	2024-07-11
Revision Received:	2025-03-18
Editorial Decision:	2025-04-14
Revision Received:	2025-04-23

Monitoring Editor: Laura Lackner

Scientific Editor: Andrea Marat

Transaction Report:

DOI: <https://doi.org/10.1083/jcb.202405162>

July 11, 2024

Re: JCB manuscript #202405162

Dr. Amit S Joshi
University of Tennessee at Knoxville
1311 Cumberland Avenue
411 Mossman Building
Knoxville 37916

Dear Dr. Joshi,

Thank you for submitting your manuscript entitled "Phosphatidic acid drives spatiotemporal distribution of Pex30 at ER-LD contact sites". The manuscript was assessed by expert reviewers, whose comments are appended to this letter. We invite you to submit a revision if you can address the reviewers' key concerns, as outlined here.

As you will see in the reviewer comments, both reviewers agree that this is an important study that provides insight into the mechanism by which Pex30 is recruited to ER-LD contact sites. The reviewers agree that the genetic and cell biological experiments are well done. They have a few recommendations to further strengthen or extend findings from these data that should be experimentally addressed in the revision. Specifically, experiments to further examine the roles of PA versus DAG (R1 #1, R2 #8) and the link between PC and PA (R2, #7) should be attempted. In addition, the levels of Pex30 and Pex30 mutants in various experiments need to be determined for the reasons articulated by Reviewer 2. For the in vitro experiments, we agree with the concerns raised by both reviewers (R1 #2, R2 #6) about the quality of data in Figure 4C, which is used to support the conclusion that Pex30 specifically binds monounsaturated PA. A revision should include dot blots of better quality or liposome floats instead of the dot blots as well as quantification of the data. In addition, using Pex30 mutations to test predictions from the MD simulations in cells as well as biochemically (R1 #3) would further strengthen the study. We feel that examining the roles of the Pex30 DysF domain in membrane curvature (R1 #4) and of Pex30 and PA in nuclear LDs (R2 #8) are important and logical avenues to pursue but are beyond the scope of the present study.

GENERAL GUIDELINES:

Text limits: Character count for an Article is < 40,000, not including spaces. Count includes title page, abstract, introduction, results, discussion, and acknowledgments. Count does not include materials and methods, figure legends, references, tables, or supplemental legends.

Figures: Articles may have up to 10 main text figures. Figures must be prepared according to the policies outlined in our Instructions to Authors, under Data Presentation, <https://jcb.rupress.org/site/misc/ifora.xhtml>. All figures in accepted manuscripts will be screened prior to publication.

Supplemental information: There are strict limits on the allowable amount of supplemental data. Articles may have up to 5 supplemental figures. Up to 10 supplemental videos or flash animations are allowed. A summary of all supplemental material should appear at the end of the Materials and methods section.

Please note that JCB now requires authors to submit Source Data used to generate figures containing gels and Western blots with all revised manuscripts. This Source Data consists of fully uncropped and unprocessed images for each gel/blot displayed in the main and supplemental figures. Since your paper includes cropped gel and/or blot images, please be sure to provide one Source Data file for each figure that contains gels and/or blots along with your revised manuscript files. File names for Source Data figures should be alphanumeric without any spaces or special characters (i.e., SourceDataF#, where F# refers to the associated main figure number or SourceDataFS# for those associated with Supplementary figures). The lanes of the gels/blots should be labeled as they are in the associated figure, the place where cropping was applied should be marked (with a box), and molecular weight/size standards should be labeled wherever possible. Source Data files will be made available to reviewers during evaluation of revised manuscripts and, if your paper is eventually published in JCB, the files will be directly linked to specific figures in the published article.

The typical timeframe for revisions is three to four months. While most universities and institutes have reopened labs and allowed researchers to begin working at nearly pre-pandemic levels, we at JCB realize that the lingering effects of the COVID-19 pandemic may still be impacting some aspects of your work, including the acquisition of equipment and reagents. Therefore, if you anticipate any difficulties in meeting this aforementioned revision time limit, please contact us and we can work with you to find an appropriate time frame for resubmission. Please note that papers are generally considered through only one revision cycle, so any revised manuscript will likely be either accepted or rejected.

Thank you for this interesting contribution to Journal of Cell Biology. You can contact us at the journal office with any questions at cellbio@rockefeller.edu.

Sincerely,

Laura Lackner, PhD
Monitoring Editor

Andrea L. Marat, PhD
Deputy Editor

Journal of Cell Biology

Reviewer #1 (Comments to the Authors (Required)):

This study looks at how peroxisome protein Pex30 influences ER subdomains and regulates ER-LD contacts. Pex30 has previously been observed at ER subdomains, and this study suggests these ER regions are enriched in phosphatidic acid. The Dysferlin domain DysF within Pex30 is also found sufficient to associate with PA. Deletion of seipin (*sei1*) elevates Pex30 and PA accumulation in ER subdomain foci, and these subdomains are specific and do not contain other proteins like Rtn1. PA levels are also elevated if Pex30 is deleted. Use of LD inducible yeast lines also indicate that Pex30 accumulates at LD formation sites with PA after the initial nascent LD formation. Deletion of the DysF domain region blunts this recruitment. In vitro, the DysF region binds a variety of charged phospholipids and can bind liposomes, and is suggested to bind specifically to monounsaturated PA species. Pex30 membrane recruitment by PA is supported by molecular dynamics simulations.

This is an interesting study that provides some new insights into how Pex30 localizes to ER subdomains, and how its loss alters ER homeostasis. The genetic and cell biological experiments are very well conducted, whereas the in vitro biochemical work remains preliminary and of variable quality. The molecular dynamics simulations may strengthen the work, but require further investigation and testing. Understanding how Pex30 and the DysF domain interacts with PA and membranes is very important, but further work and more controls are needed to further understand this and strengthen this study.

- 1) Previous work suggests Pex30 accumulates at DAG enriched ER subdomains (Joshi 2018). Understanding whether DAG or PA (or both) are the primary drivers of Pex30 recruitment to ER subdomains is important. Further experiments that delineate these possibilities will significantly strengthen this study
- 2) The dot blots presented in Fig 4C are difficult to interpret and should be redone and quantified.
- 3) The molecular dynamics simulations suggest residues and regions critical to membrane/PA binding. Further testing these MD predictions with mutations in yeast and in vitro lipid binding assays will significantly improve the study.
- 4) The MD simulations suggest the DysF region penetrates and inserts into the membrane. This may induce local membrane curvature, or alternatively that the domain senses regions of membrane curvature. Testing this further by determining if the protein binds highly curved membranes preferentially, or that it can induce membrane curvature, are important to understand.

Reviewer #2 (Comments to the Authors (Required)):

The work by House and colleagues provides insight into the molecular mechanism for the recruitment of Pex30 at the ER-LD membrane contact sites. The authors propose a model where PA recruits Pex30 at ER subdomains by binding to the dysferlin (DysF) domain of Pex30. They take advantage of a phenotype displayed by cells lacking seipin (*sei1delta*) where Pex30 (fused to 2xmCherry in the carboxy-end) displays an abnormal distribution forming large punctae which they show colocalizes with Opi1-GFP (as PA-sensor) and ER-LD sites. In dissecting which domain of Pex30 is responsible for this phenotype, the authors identify the DysF domain as a PA binding region. They provide evidence using lipid overlay and liposome floatation assays as well as MD simulations. In addition, a link to PC metabolism is also investigated.

Overall, the manuscript is well written and easy to follow and the Science is of high quality. I have some concerns for the authors to tackle in order to make sure some conclusions are correct. I have also noted that the authors failed to mention a publication from 2023 (doi: 10.1007/s12013-022-01122-z) where the role of the different Pex30 domains were investigated in relation to peroxisome biogenesis.

Specific points:

1- In a recent 2023 paper by Deori et al (doi: 10.1007/s12013-022-01122-z) a similar phenotype where Pex30-GFP forms bright punctae was shown in the presence of Sei1. I understand the authors of that study used an overexpression system, suggesting excess Pex30 may be responsible for the phenotype. Therefore, will be important to show if the levels of Pex30 in *sei1delta* cells are higher compared to wt.

Results from Deori et al should be discussed by the authors in this study.

2- Opi1 puncta seems to be nuclear in Figure 1 D *sei1delta*. Could this be checked?

3- Was the experiment in Figure 2A performed in the absence of inositol? Why is Opi1 localized at the NE at time zero?

4- Western blots showing expression of all Pex30 mutants must be included in Figure 3. This is especially important for those mutants which do not revert growth at 37 degrees.

5- Please add the amino acids positions delimiting each domain present in the mutants depicted in Figure 3A. Do all proteins have the same length?

6- The quality of the lipid overlay assay in Figure 4C must be increased and quantified

7- An obvious link between PC and PA is the step catalyzed by Spo14, which I think should be challenged experimentally.

8- Pct1 was shown to bind to nuclear LDs in Grippa et al (doi: 10.1083/jcb.201502070). Investigating the impact of Pex30 and PA in nuclear LDs will be of value in this paper.

9- Was DAG measured in the lipidomics analysis? The relevance of the PA/DAG ratio in the recruitment of Pex30 could be tested (or at least discussed).

Responses to the editor and the reviewers.

We would like to thank the reviewers and editor for their insightful, and constructive comments. In addressing their concerns, we feel the manuscript has been significantly improved.

Response to Editor's comments-

*As you will see in the reviewer comments, both reviewers agree that this is an important study that provides insight into the mechanism by which Pex30 is recruited to ER-LD contact sites. The reviewers agree that the genetic and cell biological experiments are well done. They have a few recommendations to further strengthen or extend findings from these data that should be experimentally addressed in the revision. **Specifically, experiments to further examine the roles of PA versus DAG (R1 #1, R2 #8) and the link between PC and PA (R2, #7) should be attempted. In addition, the levels of Pex30 and Pex30 mutants in various experiments need to be determined for the reasons articulated by Reviewer 2. For the in vitro experiments, we agree with the concerns raised by both reviewers (R1 #2, R2 #6) about the quality of data in Figure 4C, which is used to support the conclusion that Pex30 specifically binds monounsaturated PA. A revision should include dot blots of better quality or liposome floats instead of the dot blots as well as quantification of the data. In addition, using Pex30 mutations to test predictions from the MD simulations in cells as well as biochemically (R1 #3) would further strengthen the study. We feel that examining the roles of the Pex30 DysF domain in membrane curvature (R1 #4) and of Pex30 and PA in nuclear LDs (R2 #8) are important and logical avenues to pursue but are beyond the scope of the present study.***

R1 # 1 and R2 # 8-

We agree that it is important to understand whether DAG or PA recruit Pex30 to ER subdomains where new LDs will form. We thank the reviewers for pointing it out. To address this comment, we performed liposome flotation assays using purified Pex30 DysF domain with liposomes containing PA and DAG and DAG only (Fig 4C). Interestingly, the results of these experiments show that Pex30 DysF domain specifically binds liposomes containing PA and DAG but does not bind liposomes with only DAG. Thus, our data suggest that PA is the primary driver of Pex30 recruitment to ER subdomains and provides insight into the spatiotemporal recruitment of Pex30 to early LD biogenesis sites.

R2 #7-

We have addressed this comment by generating *spo14* Δ and *sei1spo14* Δ mutants endogenously expressing PEX30-2xmCherry and Opi1-GFP on a plasmid. We quantified the number of cells with PEX30-2xmCherry and Opi1-GFP accumulation. We found no change in either PEX30-2xmCherry or Opi1-GFP accumulation in *sei1spo14* Δ as

compared to *sei1Δ* (Fig. S4, D and E). These results indicate that the source of ectopic PA accumulation in *sei1Δ* is not PC.

R1 #2, and R2 #6-

To address this, we have replaced the dot blots in Fig. 4C with liposome flotation assays (Fig S2 F). We find that Pex30 dysferlin domain does not have a preference to monounsaturated PA over saturated PA. We also find that the domain does not have a preference to PA tail length as both 18:0/18:0 and 16:0/16:0 species of PA bound to the Pex30 DysF domain.

R1 #3-

To address this comment, we have deleted the two regions of Pex30 dysferlin domain (296-315 amino acids) and (378-398 amino acids) that are predicted to bind membrane. We expressed the GFP tagged mutant proteins in *sei1pex30Δ* cells and found that individual deletion of both regions significantly reduced the number of cells with Pex30-GFP accumulation as compared to WT. Opi1-mCherry accumulation was not affected (Fig. 5, D and E). Moreover, the mutant proteins were unable to rescue the *sei1pex30Δ* growth defect (Fig. 5F). We also confirmed that expression of Pex30 (296-315Δ) -GFP and Pex30 (378-398Δ) -GFP was not affected (Fig. 5, G and H). We attempted to purify the Pex30 DysF domain with the same deletions (296-315Δ) and (378-398Δ) and were unable to acquire pure protein following Ni-NTA affinity chromatography and size exclusion chromatography (SEC). We request to refer to the figure for reviewers included with this response. We used 5mM DTT in buffers during Ni-NTA affinity chromatography and 10mM DTT with 0.5M NaCl in PBS during SEC as an attempt to acquire pure protein. We agree that validation of these mutations *in vitro* is important, and we hope to follow up on these experiments especially to determine PA binding of residues conserved with human dysferlin domain in future studies.

R1# 4-

We thank the reviewer for this comment and agree that the dysferlin domain could be a sensor of membrane curvature or could itself induce curvature considering it binds PA a membrane curvature inducing phospholipid. Given that Pex30 contains a reticulon homology domain that is predicted to generate membrane curvature, it is likely that this domain is the primary driver of membrane curvature whereas dysferlin is the sensor. We hope to test the role of dysferlin in sensing and inducing membrane curvature in future studies.

R2# 8-

We agree with the reviewer that Pex30 and PA could be involved in nuclear LD biogenesis. Even though recent study showed that Pex30 is not associated with nuclear LD, we would like to investigate the effect of loss of Pex30 on nuclear LDs in future studies (Romanauska et al., 2024).

Response to Reviewer #1

1) Previous work suggests Pex30 accumulates at DAG enriched ER subdomains (Joshi 2018). Understanding whether DAG or PA (or both) are the primary drivers of Pex30 recruitment to ER subdomains is important. Further experiments that delineate these possibilities will significantly strengthen this study.

We agree that it is important to understand whether DAG or PA recruit Pex30 to ER subdomains where new LDs will form. We thank the reviewers for pointing it out. To address this comment, we performed liposome flotation assays using purified Pex30 DysF domain with liposomes containing PA and DAG and DAG only (Fig 4C). Interestingly, the results of these experiments show that Pex30 DysF domain specifically binds liposomes containing PA and DAG but does not bind liposomes with only DAG. Thus, our data suggest that PA is the primary driver of Pex30 recruitment to ER subdomains and provides insight into the spatiotemporal recruitment of Pex30 to early LD biogenesis sites.

2) The dot blots presented in Fig 4C are difficult to interpret and should be redone and quantified.

Point taken. To address this, we have replaced the dot blots in Fig. 4C with liposome flotation assays (Fig S2 F). We find that Pex30 dysferlin domain does not have a preference to monounsaturated PA over saturated PA. We also find that the domain does not have a preference to PA tail length as both 18:0/18:0 and 16:0/16:0 species of PA bound to the domain.

3) The molecular dynamics simulations suggest residues and regions critical to membrane/PA binding. Further testing these MD predictions with mutations in yeast and in vitro lipid binding assays will significantly improve the study.

To address this comment, we have deleted the two regions of Pex30 dysferlin domain (296-315 amino acids) and (378-398 amino acids) that are predicted to bind membrane. We expressed the GFP tagged mutant proteins in *sei1pex30Δ* cells and found that individual deletion of both regions significantly reduced the number of cells with Pex30-GFP accumulation as compared to WT. Opi1-mCherry accumulation was not affected (Fig. 5, D and E). Moreover, the mutant proteins were unable to rescue the *sei1pex30Δ* growth defect (Fig. 5F). We also confirmed that expression of Pex30 (296-315Δ) -GFP and Pex30 (378-398Δ) -GFP was not affected (Fig. 5, G and H). We attempted to purify the Pex30 DysF domain with the same deletions (296-315Δ) and (378-398Δ) and were unable to acquire pure protein following Ni-NTA affinity chromatography and size exclusion chromatography (SEC). We request to refer to the figure for reviewers included with this response. We used 5mM DTT in buffers during Ni-NTA affinity chromatography and 10mM DTT with 0.5M NaCl in PBS during SEC as an attempt to acquire pure protein. We agree that validation of these mutations *in vitro* is important, and we hope to follow up on these experiments especially to determine PA binding of residues conserved with human dysferlin domain in future studies.

4) The MD simulations suggest the DysF region penetrates and inserts into the membrane. This may induce local membrane curvature, or alternatively that the domain senses regions of membrane curvature. Testing this further by determining if the protein binds highly curved membranes preferentially, or that it can induce membrane curvature, are important to understand.

We thank the reviewer for this comment and agree that the dysferlin domain could be a sensor of membrane curvature or could itself induce curvature considering it binds PA a membrane curvature inducing phospholipid. Given that Pex30 contains a reticulon homology domain that is predicted to generate membrane curvature, it is likely that this domain is the primary driver of membrane curvature whereas dysferlin is the sensor. We hope to test the role of dysferlin in sensing and inducing membrane curvature in future studies.

Response to Reviewer # 2:

1- In a recent 2023 paper by Deori et al (doi: 10.1007/s12013-022-01122-z) a similar phenotype where Pex30-GFP forms bright punctae was shown in the presence of Sei1. I understand the authors of that study used an overexpression system, suggesting excess Pex30 may be responsible for the phenotype. Therefore, will be important to show if the levels of Pex30 in sei1delta cells are higher compared to wt. Results from Deori et al should be discussed by the authors in this study.

We agree with the reviewer that measuring Pex30 expression in sei1Δ cells is important due to the accumulation of the protein. To address this comment, we have analyzed Pex30-GFP protein levels in WT and sei1Δ cells (Fig. S1 A). We find that Pex30-GFP levels in WT and sei1Δ cells are not significantly different. These data indicate that there is a local accumulation of Pex30-GFP in sei1Δ cells rather than increased expression. We have included discussion on results from Deori et al., (Deori et al., 2022, 2023) in the introduction section.

2- Opi1 puncta seems to be nuclear in Figure 1 D sei1delta. Could this be checked?

To address this comment, we expressed Sec63-GFP, an ER marker, in sei1Δ expressing endogenously tagged Opi1-mCherry. We find that most Opi1-mCherry puncta are either on nuclear membrane or ER while very few puncta were nuclear (Fig. S1 E and F).

3- Was the experiment in Figure 2A performed in the absence of inositol? Why is Opi1 localized at the NE at time zero?

Point taken. The experiments performed in Figure 2A were performed in the presence of inositol. It is important to note that at time zero, cells were in stationary phase. Consistent with our results, Wolinsky et al. also find similar Opi1 nuclear envelope localization in the presence of inositol in stationary phase cells (Wolinski et al., 2015).

4- Western blots showing expression of all Pex30 mutants must be included in Figure 3. This is especially important for those mutants which do not revert growth at 37 degrees.

To address this comment, we have performed the recommended western blots (Fig. 3, F and G). Total cellular protein was extracted from cells expressing different GFP tagged truncations grown at 37°C. The data show no significant difference in protein expression between the WT Pex30-GFP and different Pex30-GFP truncations.

5- Please add the amino acids positions delimiting each domain present in the mutants depicted in Figure 3A. Do all proteins have the same length?

Point taken. We have included the amino acids which were deleted and have included dotted lines in the figure to show the amino acids that are absent in each mutant (Fig. 3A). All proteins do not have the same length; the shown amino acids have been deleted, and no linker region was added as a replacement.

6- The quality of the lipid overlay assay in Figure 4C must be increased and quantified

Point taken. To address this, we have replaced the dot blots in Fig. 4C with liposome flotation assays (Fig S2 F). We find that Pex30 dysferlin domain does not have a preference to monounsaturated PA over saturated PA. We also find that the domain does not have a preference to PA tail length as both 18:0/18:0 and 16:0/16:0 species of PA bound to the domain.

7- An obvious link between PC and PA is the step catalyzed by Spo14, which I think should be challenged experimentally.

We have addressed this comment by generating *spo14*Δ and *sei1spo14*Δ mutants endogenously expressing Pex30-2xmCherry and Opi1-GFP on a plasmid. We quantified the number of cells with Pex30-2xmCherry and Opi1-GFP accumulation. We found no change in either Pex30-2xmCherry or Opi1-GFP accumulation in *sei1spo14*Δ as compared to *sei1*Δ (Fig. S4, D and E). These results indicate that the source of ectopic PA accumulation in *sei1*Δ is not PC.

8- Pct1 was shown to bind to nuclear LDs in Grippa et al (doi: 10.1083/jcb.201502070). Investigating the impact of Pex30 and PA in nuclear LDs will be of value in this paper.

We agree with the reviewer that Pex30 and PA could be involved in nuclear LD biogenesis. Even though recent study showed that Pex30 is not associated with nuclear LD, we would like to investigate the effect of loss of Pex30 on nuclear LDs in future studies (Romanauska et al., 2024).

9- Was DAG measured in the lipidomics analysis? The relevance of the PA/DAG ratio in the recruitment of Pex30 could be tested (or at least discussed).

DAG was not measured in the lipidomic analysis. However, using liposome flotation assay, we determined if DysF domain also binds DAG. We demonstrate that DysF domain specifically binds PA and not DAG (Fig. 4C). Thus, our findings show that PA is the primary driver of Pex30 recruitment to ER subdomains.

References:

- Deori, N.M., T. Infant, P.K. Sundaravadivelu, R.P. Thummer, and S. Nagotu. 2022. Pex30 undergoes phosphorylation and regulates peroxisome number in *Saccharomyces cerevisiae*. *Mol. Genet. Genomics*. 297:573–590. doi:10.1007/s00438-022-01872-8.
- Deori, N.M., T. Infant, R.P. Thummer, and S. Nagotu. 2023. Characterization of the Multiple Domains of Pex30 Involved in Subcellular Localization of the Protein and Regulation of Peroxisome Number. *Cell Biochem. Biophys*. 81:39–47. doi:10.1007/s12013-022-01122-z.
- Romanauska, A., E. Stankunas, M. Schuldiner, and A. Köhler. 2024. Seipin governs phosphatidic acid homeostasis at the inner nuclear membrane. *Nat. Commun*. 15:10486. doi:10.1038/s41467-024-54811-z.
- Wolinski, H., H.F. Hofbauer, K. Hellauer, A. Cristobal-Sarramian, D. Kolb, M. Radulovic, O.L. Knittelfelder, G.N. Rechberger, and S.D. Kohlwein. 2015. Seipin is involved in the regulation of phosphatidic acid metabolism at a subdomain of the nuclear envelope in yeast. *Biochim. Biophys. Acta*. 1851:1450–64. doi:10.1016/j.bbaliip.2015.08.003.

April 14, 2025

RE: JCB Manuscript #202405162R

Amit Joshi
University of Tennessee at Knoxville

Dear Dr. Joshi:

Thank you for submitting your revised manuscript entitled "Phosphatidic acid drives spatiotemporal distribution of Pex30 at ER-LD contact sites". We would be happy to publish your paper in JCB pending final revisions necessary to meet our formatting guidelines (see details below).

A. MANUSCRIPT ORGANIZATION AND FORMATTING:

- 1) Text limits: Character count for Articles is < 40,000, not including spaces. Count includes abstract, introduction, results, discussion, and acknowledgments. Count does not include title page, figure legends, materials and methods, references, tables, or supplemental legends.
- 2) Figures limits: Articles may have up to 10 main text figures.
- 3) Figure formatting: Scale bars must be present on all microscopy images, including inset magnifications. * Molecular weight or nucleic acid size markers must be included on all gel electrophoresis. Aspect ratios of images may not be altered.
- 4) Statistical analysis: Error bars on graphic representations of numerical data must be clearly described in the figure legend. The number of independent data points (n) represented in a graph must be indicated in the legend. Statistical methods should be explained in full in the materials and methods. For figures presenting pooled data the statistical measure should be defined in the figure legends. Please also be sure to indicate the statistical tests used in each of your experiments (either in the figure legend itself or in a separate methods section) as well as the parameters of the test (for example, if you ran a t-test, please indicate if it was one- or two-sided, etc.). Also, if you used parametric tests, please indicate if the data distribution was tested for normality (and if so, how). If not, you must state something to the effect that "Data distribution was assumed to be normal but this was not formally tested."
- 5) Abstract and title: The abstract should be no longer than 160 words and should communicate the significance of the paper for a general audience. The title should be less than 100 characters including spaces. Make the title concise but accessible to a general readership.
- 6) Materials and methods: Should be comprehensive and not simply reference a previous publication for details on how an experiment was performed. Please provide full descriptions in the text for readers who may not have access to referenced manuscripts.
- 7) All antibodies, cell lines, animals, and tools used in the manuscript should be described in full, including accession numbers for materials available in a public repository such as the Resource Identification Portal. Please be sure to provide the sequences for all of your primers/oligos and RNAi constructs in the materials and methods. You must also indicate in the methods the source, species, and catalog numbers (where appropriate) for all of your antibodies. Please also indicate the acquisition and quantification methods for immunoblotting/western blots.
- 8) Microscope image acquisition: The following information must be provided about the acquisition and processing of images:
 - a. Make and model of microscope
 - b. Type, magnification, and numerical aperture of the objective lenses
 - c. Temperature
 - d. Imaging medium
 - e. Fluorochromes
 - f. Camera make and model
 - g. Acquisition software
 - h. Any software used for image processing subsequent to data acquisition. Please include details and types of operations involved (e.g., type of deconvolution, 3D reconstitutions, surface or volume rendering, gamma adjustments, etc.).

10) Supplemental materials: There are strict limits on the allowable amount of supplemental data. Articles may have up to 5 supplemental figures. Please also note that tables, like figures, should be provided as individual, editable files. A summary of all supplemental material should appear at the end of the Materials and methods section.

13) ORCID IDs: ORCID IDs are unique identifiers allowing researchers to create a record of their various scholarly contributions in a single place. Please note that ORCID IDs are now *required* for all authors. At resubmission of your final files, please be sure to provide your ORCID ID and those of all co-authors.

Please note that JCB now requires authors to submit Source Data used to generate figures containing gels and Western blots with all revised manuscripts. This Source Data consists of fully uncropped and unprocessed images for each gel/blot displayed in the main and supplemental figures. For assays performed using capillary electrophoresis and/or immunoassay-based detection, authors should instead provide the electropherogram graph(s) for each experiment, plotting fluorescence/chemiluminescence intensity vs. molecular weight/size. Please be sure to provide one Source Data file for each figure gels, blots, and/or capillary electrophoresis assays along with your revised manuscript files. File names for Source Data figures should be alphanumeric without any spaces or special characters (i.e., SourceDataF#, where F# refers to the associated main figure number or SourceDataFS# for those associated with Supplementary figures). For traditional gels and blots, the lanes of the gels/blots should be labeled as they are in the associated figure, the place where cropping was applied should be marked (with a box), and molecular weight/size standards should be labeled wherever possible. For capillary electrophoresis assays, each trace in the graph should be color-coded and labeled to indicate which protein, gene, or sample is being measured (please try to avoid red/green combinations to accommodate our color-blind readers).

Journal of Cell Biology now requires a data availability statement for all research article submissions. These statements will be published in the article directly above the Acknowledgments. The statement should address all data underlying the research presented in the manuscript. Please visit the JCB instructions for authors for guidelines and examples of statements at (<https://rupress.org/jcb/pages/editorial-policies#data-availability-statement>).

B. FINAL FILES:

****It is JCB policy that if requested, original data images must be made available to the editors. Failure to provide original images**

upon request will result in unavoidable delays in publication. Please ensure that you have access to all original data images prior to final submission.**

Thank you for your attention to these final processing requirements. Please revise and format the manuscript and upload materials within 7 days. If you need an extension for whatever reason, please let us know and we can work with you to determine a suitable revision period.

Thank you for this interesting contribution, we look forward to publishing your paper in Journal of Cell Biology.

Sincerely,

Laura Lackner, PhD
Monitoring Editor

Andrea L. Marat, PhD
Deputy Editor

Journal of Cell Biology

Reviewer #1 (Comments to the Authors (Required)):

The revision addresses the major concerns. Liposome binding assays have now been conducted replacing lipid dot blots. Additional Pex30 DysF domain mutants are also analyzed. Specific imaging experiments have also been conducted to address more minor points. This is an important study that should be published.

Reviewer #2 (Comments to the Authors (Required)):

The authors have addressed all my concerns and I think new experimental additions have greatly improved the paper.